# Label-Guided Representation Learning for Incomplete Multi-View Multi-Label Classification

Yang Li [* 1]    Quanjiang Li [* 1]    Tingjin Luo [1]

## Abstract

Incomplete multi-view multi-label classification addresses scenarios where views and labels are partially missing. While existing methods treat labels solely as supervision signals, they overlook the semantic structure inherent in partial annotations. To tackle this problem, we propose Label-Guided Representation Learning (LGRL) to systematically exploit label semantics as structural prior information. Specifically, we construct a semantic-aware mixture prior via learnable category prototypes to explicitly guide representation extraction across views. Furthermore, category-specific conditional posteriors are introduced to leverage these prototypes as Bayesian experts to govern multi-view fusion. Besides, we further derive a principled label-driven information bottleneck objective balancing reconstruction sufficiency with cross-view consistency, enabling category-conditional reasoning. Extensive experimental results demonstrate the effectiveness of LGRL across benchmark datasets as well as applications in sports analytics and medical imaging.

## 1. Introduction

Multi-view learning exploits complementary information from heterogeneous data sources to capture richer semantic representations than single-view counterparts (Pan & Kang, 2021; Yang et al., 2021; Yuan et al., 2025). Multi-label classification characterizes objects with multiple semantic attributes simultaneously (Si et al., 2023). Real-world applications frequently exhibit both characteristics: a news article can be described by textual content, associated images, and user comments, while belonging to multiple categories such as politics, economy, and society (Ye et al., 2024; Chen et al.,

2024). The confluence of these two paradigms gives rise to multi-view multi-label classification (MvMLC). In practice, acquiring complete multi-view multi-label data is often prohibitively expensive or technically infeasible. Sensor failures, privacy constraints, and annotation costs frequently result in missing views or incomplete label sets (Wang et al., 2025; Tan et al., 2018; Luo et al., 2025). The resulting dual incompleteness poses significant challenges: missing views disrupt cross-view dependency modeling, while partial labels provide insufficient supervision and obscure label correlation structures. Consequently, incomplete multi-view multi-label classification (iMvMLC) has attracted more attention as it addresses the general scenarios in real world.

Existing iMvMLC methods have advanced along three main directions. The first line focuses on robust feature extraction under view incompleteness. Early matrix factorization approaches such as iMVWL (Tan et al., 2018) learn shared subspaces robust to missing views, while recent deep methods including DICNet (Liu et al., 2023b) employ instance-level contrastive learning to enhance consensus representations. The second line addresses multi-view fusion strategies. LMVCAT (Liu et al., 2023c) leverages transformer architectures for cross-view aggregation, MFD (Xie et al., 2025) factorizes representations into view-consistent and view-specific components, SIP (Liu et al., 2024b) formulates fusion through an information bottleneck lens, and URDF (Wen et al., 2026) introduces uncertainty-driven dynamic fusion to estimate per-sample feature reliability for more reliable view aggregation. The third line explores label semantic modeling. NAIM3L (Li & Chen, 2021) combines global high-rank and local low-rank constraints to capture label correlations, CSA (Shao & Xu, 2025) proposes enhanced contrastive learning for cross-view semantic alignment, SSP (Li et al., 2024) employs graph constraint learning to preserve semantic structure consistency among samples during feature extraction, and TDLSR (Li et al., 2025) constructs view-specific sample topology and prototype association graphs to derive label-specific representations with theoretical generalization guarantees.

Despite the progress, existing methods have not fully exploited label semantics, and most approaches still treat labels merely as supervision signals at the output layer. We identify three limitations at progressive levels. First, at

---

[*]Equal contribution [1]College of Science, National University of Defense Technology, Changsha, China. Correspondence to: Tingjin Luo <tingjinluo@hotmail.com>.

*Proceedings of the 43rd International Conference on Machine Learning*, Seoul, South Korea. PMLR 306, 2026. Copyright 2026 by the author(s).

the label level, partial annotations encode meaningful co-occurrence patterns, e.g., "beach" increases the likelihood of "ocean" while decreasing that of "mountain", yet such inter-label semantic structures are seldom explicitly modeled and exploited. Second, at the representation level, existing methods employ label information only as supervision signals at the output layer through classification losses, while the latent representation learning itself remains label-agnostic, resulting in representations that may preserve input information well but lack explicit semantic structure aligned with label relationships. Third, at the fusion level, current mechanisms aggregate multi-view information into a single shared representation (Wen et al., 2023; Tan et al., 2024; Liu et al., 2024b), implicitly assuming all categories benefit equally from the same feature combination. In practice, different categories often depend on distinct view contributions, and a category-agnostic fusion approach typically fails to capture such heterogeneity, thereby producing suboptimal representations (Zhang et al., 2024).

To address these limitations, we propose Label-Guided Representation Learning (LGRL), a unified framework that systematically leverages label semantics to guide representation extraction and multi-view fusion. For label-guided representation extraction, we derive a label-aware information bottleneck objective that balances reconstruction sufficiency with cross-view consistency, ensuring that the learned representations retain task-relevant information under the guidance of label supervision. To further encode label semantics into the latent space, we introduce a semantic-informed mixture prior with learnable category prototypes, where samples are encouraged to align with positive prototypes while being separated from negative ones through contrastive learning, yielding a geometrically structured latent space that captures label co-occurrence patterns. For label-guided multi-view fusion, we introduce category-specific conditional posteriors, in which each prototype serves as a precision-weighted expert within a Product-of-Experts (PoE) framework. This design enables hypothesis-driven multi-label reasoning, adaptively modulating view contributions in accordance with category-specific semantic cues. Finally, we evaluate the effectiveness of LGRL on six benchmarks and scenarios including NBA player analysis and chest radiograph screening. Our main contributions are summarized as follows:

- We propose LGRL, a unified framework that systematically exploits label semantics to guide both representation extraction and multi-view fusion for iMvMLC.

- We derive a label-aware information bottleneck objective and introduce learnable category prototypes that impose semantic structure on the latent space via contrastive alignment, enabling discriminative representations that capture label co-occurrence patterns.

- We design category-specific conditional posteriors

where prototypes serve as Bayesian experts, enabling hypothesis-based reasoning that adapts fusion weights to each category's semantic characteristics.

- Extensive experiments demonstrate that LGRL consistently outperforms state-of-the-art methods across various missing ratios.

## 2. Method

As discussed in Section 1, existing iMvMLC methods under-utilize the semantic information contained in partial annotations. We propose Label-Guided Representation Learning (LGRL) to incorporate label semantics into both representation learning and multi-view fusion. As illustrated in Figure 1, LGRL comprises three components: (1) a semantic-informed mixture prior that encodes label semantics into latent space geometry via learnable category prototypes; (2) a task-relevant learning objective balancing reconstruction sufficiency with cross-view consistency; and (3) a Bayesian fusion mechanism where prototypes serve as experts for hypothesis-based reasoning.

### 2.1. Problem Formulation

Consider a dataset $\mathcal{D} = \{(\{\mathbf{x}_i^{(v)}\}_{v=1}^m, \mathbf{y}_i)\}_{i=1}^n$ comprising $n$ samples, where each sample $i$ is associated with $m$ views and a multi-label annotation. The $v$-th view is represented as $\mathbf{x}_i^{(v)} \in \mathbb{R}^{d_v}$, and the label vector $\mathbf{y}_i \in \{0,1\}^C$ encodes membership across $C$ categories.

In practical scenarios, both views and labels suffer from incompleteness. Let $\mathcal{V}_i \subseteq \{1, \ldots, m\}$ denote the set of available views for sample $i$. For labels, let $\mathcal{O}_i \subseteq \{1, \ldots, C\}$ denote the observed set comprising both positive and negative annotations, with $\mathcal{U}_i = \{1, \ldots, C\} \setminus \mathcal{O}_i$ the unobserved complement. Given incomplete observations $\{(\{\mathbf{x}_i^{(v)}\}_{v \in \mathcal{V}_i}, \{y_{i,c}\}_{c \in \mathcal{O}_i})\}_{i=1}^n$, the goal is to learn a shared representation $\mathbf{z}_i \in \mathbb{R}^{d_z}$ that is robust to missing views and enables accurate prediction of the complete label vector $\mathbf{y}_i$.

### 2.2. Semantic-Informed Mixture Prior

Standard variational frameworks adopt an uninformative prior $p(\mathbf{z}) = \mathcal{N}(\mathbf{0}, \mathbf{I})$, treating the latent space as semantically unstructured. We argue that even partial label annotations reveal valuable structure, including label co-occurrence patterns and inter-class relationships that should inform representation geometry. These observations motivate a semantic-informed mixture prior that explicitly encodes such structure.

Each category $c \in \{1, \ldots, C\}$ is associated with a dedicated learnable prototype distribution, which is

$$p(\mathbf{z} \mid y_c = 1) = \mathcal{N}\big(\boldsymbol{\mu}_c, \mathrm{diag}(\boldsymbol{\sigma}_c^2)\big). \tag{1}$$

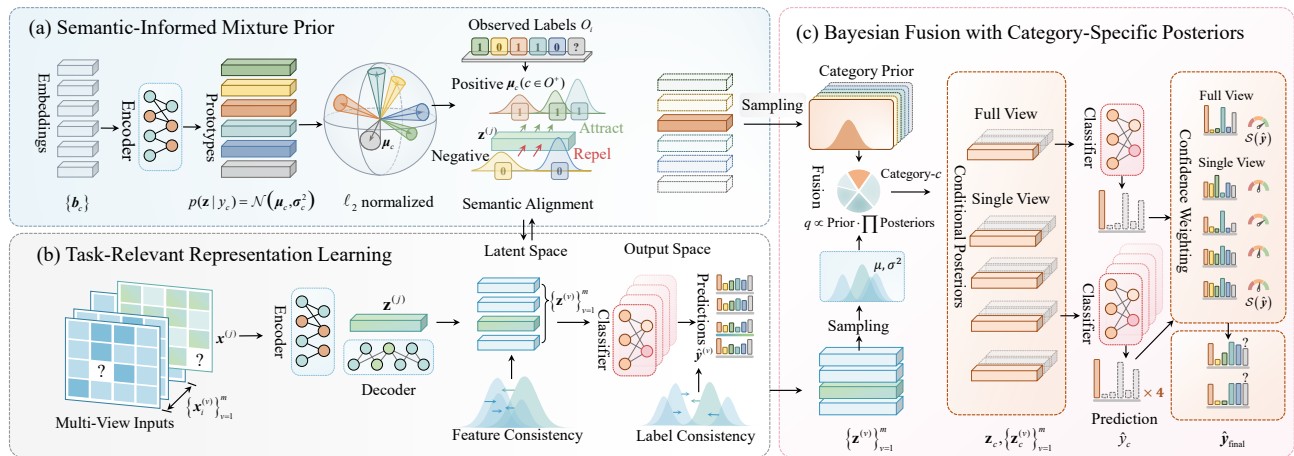

*Figure 1.* Overview of LGRL. (a) Semantic-Informed Mixture Prior encodes label relationships via learnable category prototypes. (b) Task-Relevant Representation Learning enforces consistency at both feature and label levels. (c) Bayesian Fusion integrates category priors into posterior computation for hypothesis-based prediction.

The prototype parameters are generated by a dedicated Label Encoder from trainable category embeddings $\{\mathbf{b}_c \in \mathbb{R}^{d_b}\}_{c=1}^{C}$. Each embedding $\mathbf{b}_c$ is initialized as a one-hot vector and transformed through two separate two-layer MLPs to produce $\boldsymbol{\mu}_c = h_\mu(\mathbf{b}_c)$ and $\boldsymbol{\sigma}_c^2 = \text{softplus}(h_\sigma(\mathbf{b}_c))$, where $h_\mu, h_\sigma : \mathbb{R}^{d_b} \to \mathbb{R}^{d_z}$. As training progresses, the embeddings gradually evolve to capture label correlations from joint optimization with observed labels.

For a sample with observed labels $\mathcal{O}$, its latent representation is aligned with the semantic structure via contrastive learning. Specifically, positive labels indicate categories whose prototypes should attract the learned representation, while negative labels indicate categories whose prototypes should repel it. Operating on $\ell_2$-normalized representations $\tilde{\mathbf{z}} = \mathbf{z}/\|\mathbf{z}\|$ and prototypes $\tilde{\boldsymbol{\mu}}_c = \boldsymbol{\mu}_c/\|\boldsymbol{\mu}_c\|$, the semantic alignment loss is then defined as

$$\mathcal{L}_{\text{SA}} = - \sum_{c \in \mathcal{O}^+} \log \frac{\exp\big(\tilde{\mathbf{z}}^\top \tilde{\boldsymbol{\mu}}_c / \tau\big)}{\sum_{j \in \mathcal{O}} \exp\big(\tilde{\mathbf{z}}^\top \tilde{\boldsymbol{\mu}}_j / \tau\big)}, \qquad (2)$$

where $\mathcal{O}^+$ denotes positive labels within $\mathcal{O}$ and $\tau$ is a temperature parameter. Minimizing Eq. (2) organizes the latent space so that samples with similar labels cluster around shared prototypes, while samples with different labels are pushed apart from it.

### 2.3. Task-Relevant Representation Learning

With the structured prior established, we formulate the learning objective by balancing two desiderata, namely *sufficiency* that retains task-relevant information for reconstruction and classification, and *consistency* that enforces cross-view agreement. While existing information bottleneck approaches (Liu et al., 2024b) optimize a unified objective

over shared information, we decompose it into feature-level and label-level components, enabling independent control over reconstruction quality and classification performance.

For feature-level learning, the sufficiency term $I(\mathbf{x}^{(v)}; \mathbf{z}^{(v)})$ preserves information for reconstruction, while the consistency term $I(\mathbf{x}^{\sim(v)}; \mathbf{z}^{(v)} \mid \mathbf{x}^{(v)})$ suppresses view-specific artifacts. The objective is

$$\mathcal{I}_{\text{feat}}^{(v)} = \underbrace{I(\mathbf{x}^{(v)}; \mathbf{z}^{(v)})}_{\text{Sufficiency}} - \alpha \underbrace{I(\mathbf{x}^{\sim(v)}; \mathbf{z}^{(v)} \mid \mathbf{x}^{(v)})}_{\text{Consistency}}, \qquad (3)$$

where $\mathbf{x}^{\sim(v)}$ denotes all views except view $v$.

Since feature-level learning may preserve low-level details while discarding semantic patterns crucial for classification, we incorporate label information. The sufficiency term $I(\mathbf{y}; \mathbf{z}^{(v)})$ ensures predictive capacity, while the consistency term $I(\mathbf{y}; \mathbf{z}^{(v)} \mid \mathbf{z}^{\sim(v)})$ enforces concordant evidence across views. The label-level information objective can be formulated as

$$\mathcal{I}_{\text{label}}^{(v)} = \underbrace{I(\mathbf{y}; \mathbf{z}^{(v)})}_{\text{Sufficiency}} - \gamma \underbrace{I(\mathbf{y}; \mathbf{z}^{(v)} \mid \mathbf{z}^{\sim(v)})}_{\text{Consistency}}, \qquad (4)$$

where $\mathbf{z}^{\sim(v)}$ denotes latent representations from all views except $v$. The unified objective sums over all observed views

$$\max \sum_{v \in \mathcal{V}} \Big[ \mathcal{I}_{\text{feat}}^{(v)} + \lambda \cdot \mathcal{I}_{\text{label}}^{(v)} \Big], \qquad (5)$$

where $\lambda$ balances feature and label information.

Direct optimization of mutual information is intractable. Variational bounds are derived, specifically lower bounds for sufficiency terms and upper bounds for consistency terms,

with complete derivations in Appendix A. For reconstruction, introducing a variational decoder $p_\theta(\mathbf{x}^{(v)} \mid \mathbf{z}^{(\mathbf{v})})$ yields

$$I(\mathbf{x}^{(v)}; \mathbf{z}^{(v)})$$
$$\geq \mathbb{E}_{p(\mathbf{x}^{(v)})} \mathbb{E}_{q_\phi(\mathbf{z}^{(v)} \mid \mathbf{x}^{(v)})} \left[ \log p_\theta(\mathbf{x}^{(v)} \mid \mathbf{z}^{(v)}) \right]. \quad (6)$$

For encoder consistency, the fused representation $\mathbf{z}$ aggregates more information about other views than $\mathbf{z}^{(v)}$ alone, motivating the upper bound

$$I(\mathbf{x}^{\sim(v)}; \mathbf{z}^{(v)} \mid \mathbf{x}^{(v)})$$
$$\leq \mathbb{E}_{p(\{\mathbf{x}\})} \left[ D_{\mathrm{KL}} \left( q(\mathbf{z} \mid \{\mathbf{x}\}) \,\|\, q_\phi(\mathbf{z} \mid \mathbf{x}^{(v)}) \right) \right], \quad (7)$$

where $q(\mathbf{z} \mid \{\mathbf{x}\})$ is the fused posterior via Product-of-Experts (Section 2.4).

Following similar reasoning, bounds for label-related terms are derived by introducing a classifier $p_\psi(\mathbf{y} \mid \mathbf{z})$. The label sufficiency bound is

$$I(\mathbf{y}; \mathbf{z}^{(v)}) \geq \mathbb{E}_{p(\mathbf{x}^{(v)})} \mathbb{E}_{q_\phi(\mathbf{z}^{(v)} \mid \mathbf{x}^{(v)})} \left[ \log p_\psi(\mathbf{y} \mid \mathbf{z}^{(v)}) \right]. \quad (8)$$

For label consistency, comparing predictions from fused and view-specific representations yields

$$I(\mathbf{y}; \mathbf{z}^{(v)} \mid \mathbf{z}^{\sim(v)})$$
$$\leq \mathbb{E}_{p(\{\mathbf{x}\})} \mathbb{E}_{q(\mathbf{z} \mid \{\mathbf{x}\})} \left[ D_{\mathrm{KL}} \left( p_\psi(\mathbf{y} \mid \mathbf{z}) \,\|\, p_\psi(\mathbf{y} \mid \mathbf{z}^{(v)}) \right) \right]. \quad (9)$$

Combining Eqs.(6)-(9) and converting to minimization, the feature-level loss is

$$\mathcal{L}_{\mathrm{feat}}^{(v)} = \underbrace{-\mathbb{E}_{q_\phi(\mathbf{z} \mid \mathbf{x}^{(v)})}[\log p_\theta(\mathbf{x}^{(v)} \mid \mathbf{z})]}_{\text{Reconstruction}}$$
$$+ \alpha \underbrace{D_{\mathrm{KL}}(q(\mathbf{z} \mid \{\mathbf{x}\}) \| q_\phi(\mathbf{z} \mid \mathbf{x}^{(v)}))}_{\text{Encoder Consistency}}, \quad (10)$$

and the label-level loss is

$$\mathcal{L}_{\mathrm{label}}^{(v)} = \underbrace{-\mathbb{E}_{q_\phi(\mathbf{z} \mid \mathbf{x}^{(v)})}[\log p_\psi(\mathbf{y} \mid \mathbf{z})]}_{\text{Classification}}$$
$$+ \gamma \underbrace{\mathbb{E}_{q(\mathbf{z} \mid \{\mathbf{x}\})}[D_{\mathrm{KL}}(p_\psi(\mathbf{y} \mid \mathbf{z}) \| p_\psi(\mathbf{y} \mid \mathbf{z}^{(v)}))]}_{\text{Classifier Consistency}}. \quad (11)$$

The task-relevant objective integrates Eqs. (10) and (11) across all available views

$$\mathcal{L}_{\mathrm{TR}} = \sum_{v \in \mathcal{V}} \left[ \mathcal{L}_{\mathrm{feat}}^{(v)} + \lambda \mathcal{L}_{\mathrm{label}}^{(v)} \right]. \quad (12)$$

### 2.4. Bayesian Fusion with Category-Specific Posteriors

A key innovation of LGRL is treating category prototypes as active participants in Bayesian inference to achieve label-guided multi-view fusion. While the mixture prior (Section 2.2) captures label co-occurrence at the representation level, different categories may rely on distinct feature combinations. For instance, detecting "texture" may emphasize edge patterns, whereas identifying "sentiment" may depend more on color distributions. Unlike approaches that aggregate multi-view information into a single shared representation (Wen et al., 2023; Tan et al., 2024), we integrate category semantics directly into fusion through category-specific conditional posteriors, enabling hypothesis-based multi-label reasoning.

For a sample with available views $\mathcal{V}$, let $q_\phi(\mathbf{z} \mid \mathbf{x}^{(v)}) = \mathcal{N}(\boldsymbol{\mu}^{(v)}, \mathrm{diag}((\boldsymbol{\sigma}^{(v)})^2))$ denote the encoder output for view $v$. For each category $c$, the conditional posterior factorizes via Product-of-Experts (PoE)

$$q(\mathbf{z} \mid \{\mathbf{x}\}, y_c = 1) \propto \underbrace{p(\mathbf{z} \mid y_c = 1)}_{\text{Category Prior}} \cdot \prod_{v \in \mathcal{V}} \underbrace{q_\phi(\mathbf{z} \mid \mathbf{x}^{(v)})}_{\text{View Posteriors}}, \quad (13)$$

where the category prior $p(\mathbf{z} \mid y_c = 1) = \mathcal{N}(\boldsymbol{\mu}_c, \mathrm{diag}(\boldsymbol{\sigma}_c^2))$ encodes the expected feature distribution for category $c$ and serves as an additional expert in fusion.

Since all components are Gaussian, the PoE yields closed-form precision-weighted parameters

$$\begin{cases} \boldsymbol{\tau}_c = \boldsymbol{\sigma}_c^{-2} + \sum_{v \in \mathcal{V}} (\boldsymbol{\sigma}^{(v)})^{-2}, \\ \boldsymbol{\mu}_c^{\mathrm{post}} = \boldsymbol{\tau}_c^{-1} \left( \boldsymbol{\sigma}_c^{-2} \boldsymbol{\mu}_c + \sum_{v \in \mathcal{V}} (\boldsymbol{\sigma}^{(v)})^{-2} \boldsymbol{\mu}^{(v)} \right), \end{cases} \quad (14)$$

where sources with lower variance (higher confidence) contribute more. The formulation naturally handles missing views, as absent views simply do not participate.

Sampling $\mathbf{z}_c \sim \mathcal{N}(\boldsymbol{\mu}_c^{\mathrm{post}}, \boldsymbol{\tau}_c^{-1})$, the hypothesis "$y_c = 1$" is evaluated via

$$p(y_c = 1 \mid \mathbf{z}_c) = \sigma(f_c(\mathbf{z}_c)), \quad (15)$$

where $f_c(\cdot)$ is a category-specific classifier. The sampled $\mathbf{z}_c$ incorporates the category-$c$ prototype as prior, biasing the representation toward category-relevant subspaces, while the classifier evaluates whether the observed evidence supports the hypothesis.

Since individual views may exhibit varying reliability across categories, single-view conditional posteriors $q(\mathbf{z} \mid \mathbf{x}^{(v)}, y_c = 1) \propto p(\mathbf{z} \mid y_c = 1) \cdot q_\phi(\mathbf{z} \mid \mathbf{x}^{(v)})$ are further considered, yielding $(1 + |\mathcal{V}|)$ prediction sources in total. The predictions are aggregated via confidence weighting with $\mathcal{S}(\hat{\mathbf{y}}) = \sum_c (\hat{y}_c^2 + (1 - \hat{y}_c)^2)$ favoring decisive predictions

$$\hat{\mathbf{y}}_{\mathrm{final}} = \sum_{k=0}^{|\mathcal{V}|} \frac{\exp(\mathcal{S}(\hat{\mathbf{y}}_k)/\kappa)}{\sum_{j=0}^{|\mathcal{V}|} \exp(\mathcal{S}(\hat{\mathbf{y}}_j)/\kappa)} \cdot \hat{\mathbf{y}}_k, \quad (16)$$

where $k = 0$ indexes the full-view posterior, $k \geq 1$ indexes single-view posteriors, and $\kappa$ controls weighting sharpness.

The aggregated predictions are supervised via binary cross-entropy over observed labels

$$\mathcal{L}_{\text{CE}} = -\frac{1}{|\mathcal{O}|} \sum_{c \in \mathcal{O}} [y_c \log \hat{y}_c + (1 - y_c) \log(1 - \hat{y}_c)]. \tag{17}$$

Finally, to establish dual robustness at both the representation level through the semantic priors and the decision level through the confidence weighting mechanism, the overall objective of LGRL is formulated as

$$\mathcal{L} = \mathcal{L}_{\text{TR}} + \beta \mathcal{L}_{\text{SA}} + \mathcal{L}_{\text{CE}}. \tag{18}$$

## 2.5. Network Architecture and Training

Each view $v$ is equipped with a dedicated encoder $E_v$ and decoder $D_v$. The encoder maps input $\mathbf{x}^{(v)} \in \mathbb{R}^{d_v}$ to $q_\phi(\mathbf{z}^{(v)} \mid \mathbf{x}^{(v)}) = \mathcal{N}(\boldsymbol{\mu}^{(v)}(\mathbf{x}^{(v)}), \text{diag}((\boldsymbol{\sigma}^{(v)}(\mathbf{x}^{(v)}))^2))$, where $\boldsymbol{\mu}^{(v)}, \boldsymbol{\sigma}^{(v)} : \mathbb{R}^{d_v} \to \mathbb{R}^{d_z}$ are neural networks. Sampling uses the reparameterization trick $\mathbf{z}^{(v)} = \boldsymbol{\mu}^{(v)}(\mathbf{x}^{(v)}) + \boldsymbol{\sigma}^{(v)}(\mathbf{x}^{(v)}) \odot \boldsymbol{\epsilon}$ with $\boldsymbol{\epsilon} \sim \mathcal{N}(\mathbf{0}, \mathbf{I})$.

The multi-view joint posterior $q(\mathbf{z} \mid \{\mathbf{x}\})$ is obtained via PoE fusion, yielding $q(\mathbf{z} \mid \{\mathbf{x}\}) = \mathcal{N}(\boldsymbol{\mu}_{\text{PoE}}, \text{diag}(\boldsymbol{\sigma}^2_{\text{PoE}}))$ with $\boldsymbol{\sigma}^{-2}_{\text{PoE}} = \sum_{v \in \mathcal{V}} (\boldsymbol{\sigma}^{(v)})^{-2} + \mathbf{1}$ and $\boldsymbol{\mu}_{\text{PoE}} = \boldsymbol{\sigma}^2_{\text{PoE}} \sum_{v \in \mathcal{V}} (\boldsymbol{\sigma}^{(v)})^{-2} \boldsymbol{\mu}^{(v)}$. Only available views contribute, naturally handling missingness. Note that this fusion adopts the uninformative prior $\mathcal{N}(\mathbf{0}, \mathbf{I})$ and is used for the consistency terms, in contrast to the prototype-conditioned fusion of Eq. (13) used for prediction. During training, category prototypes are sampled and jointly optimized; at test time, deterministic prototype means $\{\boldsymbol{\mu}_c = h_\mu(\mathbf{b}_c)\}_{c=1}^C$ are used as fixed semantic anchors. The main procedure of LGRL is summarized in Algorithm 1.

## 3. Experiments

### 3.1. Experimental Settings

**Datasets and Metrics.** We conduct experiments on six widely-used multi-view multi-label datasets: Corel5k (Duygulu et al., 2002), ESPGame (Ahn & Dabbish, 2004), IAPRTC12 (Grubinger et al., 2006), Mirflickr (Huiskes & Lew, 2008), Pascal07 (Everingham et al., 2010), and OBJECT (Hao et al., 2024). Following (Liu et al., 2023b; Wen et al., 2023), we adopt six evaluation metrics: Hamming Loss (HL), Ranking Loss (RL), OneError (OE), Coverage (Cov), Average Precision (AP), and Area Under Curve (AUC). For clearer comparison, we report the results of 1-HL, 1-OE, 1-Cov and 1-RL, where higher values indicate better performance.

**Comparison Methods.** We compare LGRL against nine

---

**Algorithm 1** Training Procedure of LGRL

---
1: **Input:** Dataset $\mathcal{D}$, the hyperparameters $\alpha, \beta, \gamma, \lambda, \kappa$
2: Initialize category embeddings $\{\mathbf{b}_c\}_{c=1}^C$
3: **for** each mini-batch $\mathcal{B} \subset \mathcal{D}$ **do**
4:     **for** each sample $i \in \mathcal{B}$ **do**
5:         **for** each $v \in \mathcal{V}_i$ **do**
6:             Encode: $(\boldsymbol{\mu}^{(v)}, (\boldsymbol{\sigma}^{(v)})^2) \leftarrow E_v(\mathbf{x}_i^{(v)})$
7:             Sample $\mathbf{z}_i^{(v)}$ via reparameterization
8:         **end for**
9:         Fuse views via PoE to obtain $q(\mathbf{z} \mid \{\mathbf{x}_i\})$
10:         **for** each category $c = 1$ **to** $C$ **do**
11:             Generate prototype:
                $(\boldsymbol{\mu}_c, \boldsymbol{\sigma}_c^2) \leftarrow (h_\mu(\mathbf{b}_c), h_\sigma(\mathbf{b}_c))$
12:             Compute conditional posterior via Eq. (13)
13:             Sample $\mathbf{z}_{i,c}$ and predict $\hat{y}_{i,c}$ via Eq. (15)
14:         **end for**
15:         Aggregate predictions via Eq. (16)
16:     **end for**
17:     Compute $\mathcal{L} = \mathcal{L}_{\text{TR}} + \beta \mathcal{L}_{\text{SA}} + \mathcal{L}_{\text{CE}}$
18:     Update parameters by minimizing $\mathcal{L}$
19: **end for**

---

state-of-the-art methods: AIMNet (Liu et al., 2024a), DICNet (Liu et al., 2023b), DIMC (Wen et al., 2023), iMVWL (Tan et al., 2018), LMVCAT (Liu et al., 2023c), MTD (Liu et al., 2023a), SIP (Liu et al., 2024b), TDLSR (Li et al., 2025), and LVSL (Zhao et al., 2022). Among these, the first eight methods can handle both missing views and labels simultaneously. For LVSL, which cannot handle missing data, we impute missing views with feature means and fill unknown labels with zeros. All baseline parameters are set to their recommended values.

**Implementation Details.** Each dataset is split into training, validation, and test sets with a ratio of 7:1:2. To simulate incomplete scenarios, we randomly mark a specified proportion of view instances as unavailable according to the Partial Example Ratio (PER), while ensuring each sample retains at least one complete view. For label incompleteness, we randomly omit both positive and negative annotations according to the Label Missing Ratio (LMR). Optimization uses Adam with learning rate $10^{-4}$ and contrastive temperature $\tau = 0.1$. All other hyperparameters ($\alpha, \gamma, \kappa, \lambda, \beta$) are fixed at 1. Experiments run on an NVIDIA RTX 4090 (24GB), with results averaged over 10 runs for the 50% PER and LMR setting, and 5 runs for other configurations.

### 3.2. Experimental Results and Analysis

To evaluate the effectiveness of LGRL in handling absent views and labels, we benchmark it against nine methods across six datasets with varying levels of data sparsity. The missing ratios for both views (PER) and labels (LMR) are

*Table 1.* Experimental results of ten methods on the six datasets with 50% PER and 50% LMR. "AVE" refers to the mean ranking of the corresponding method across all six metrics.

| DATA | METRIC | AIMNet | DICNet | DIMC | iMVWL | LMVCAT | LVSL | MTD | SIP | TDLSR | LGRL |
|---|---|---|---|---|---|---|---|---|---|---|---|
| **COR** | **1-HL** | $0.988_{0.000}$ | $0.987_{0.000}$ | $0.987_{0.000}$ | $0.978_{0.000}$ | $0.986_{0.000}$ | $0.987_{0.000}$ | $0.988_{0.000}$ | $0.988_{0.000}$ | $0.988_{0.000}$ | $0.988_{0.000}$ |
| | **1-OE** | $0.478_{0.011}$ | $0.460_{0.012}$ | $0.446_{0.009}$ | $0.308_{0.017}$ | $0.448_{0.011}$ | $0.353_{0.017}$ | $0.492_{0.011}$ | $0.492_{0.014}$ | $0.541_{0.014}$ | $0.567_{0.016}$ |
| | **1-Cov** | $0.766_{0.004}$ | $0.726_{0.007}$ | $0.709_{0.008}$ | $0.701_{0.003}$ | $0.720_{0.006}$ | $0.720_{0.005}$ | $0.754_{0.005}$ | $0.780_{0.004}$ | $0.801_{0.009}$ | $0.810_{0.004}$ |
| | **1-RL** | $0.900_{0.002}$ | $0.881_{0.004}$ | $0.874_{0.004}$ | $0.864_{0.002}$ | $0.876_{0.004}$ | $0.879_{0.003}$ | $0.893_{0.004}$ | $0.908_{0.003}$ | $0.917_{0.004}$ | $0.923_{0.002}$ |
| | **AP** | $0.404_{0.005}$ | $0.381_{0.006}$ | $0.370_{0.005}$ | $0.281_{0.005}$ | $0.379_{0.006}$ | $0.311_{0.005}$ | $0.414_{0.006}$ | $0.414_{0.006}$ | $0.450_{0.006}$ | $0.469_{0.006}$ |
| | **AUC** | $0.903_{0.002}$ | $0.883_{0.004}$ | $0.877_{0.004}$ | $0.867_{0.002}$ | $0.879_{0.003}$ | $0.882_{0.002}$ | $0.895_{0.003}$ | $0.910_{0.002}$ | $0.919_{0.004}$ | $0.925_{0.002}$ |
| | **AVE** | 3.500 | 5.000 | 7.000 | 8.500 | 6.333 | 6.500 | 3.667 | 2.667 | 1.833 | 1.000 |
| **ESP** | **1-HL** | $0.983_{0.000}$ | $0.983_{0.000}$ | $0.983_{0.000}$ | $0.972_{0.000}$ | $0.982_{0.000}$ | $0.983_{0.000}$ | $0.983_{0.000}$ | $0.983_{0.000}$ | $0.983_{0.000}$ | $0.983_{0.000}$ |
| | **1-OE** | $0.442_{0.006}$ | $0.440_{0.009}$ | $0.431_{0.009}$ | $0.343_{0.010}$ | $0.431_{0.006}$ | $0.365_{0.006}$ | $0.452_{0.007}$ | $0.450_{0.006}$ | $0.477_{0.007}$ | $0.511_{0.011}$ |
| | **1-Cov** | $0.621_{0.003}$ | $0.601_{0.003}$ | $0.586_{0.004}$ | $0.548_{0.004}$ | $0.587_{0.003}$ | $0.578_{0.002}$ | $0.617_{0.004}$ | $0.622_{0.004}$ | $0.646_{0.004}$ | $0.671_{0.004}$ |
| | **1-RL** | $0.845_{0.002}$ | $0.836_{0.002}$ | $0.830_{0.002}$ | $0.807_{0.002}$ | $0.827_{0.002}$ | $0.829_{0.001}$ | $0.843_{0.002}$ | $0.847_{0.002}$ | $0.859_{0.002}$ | $0.871_{0.002}$ |
| | **AP** | $0.305_{0.003}$ | $0.300_{0.003}$ | $0.294_{0.003}$ | $0.243_{0.004}$ | $0.293_{0.003}$ | $0.266_{0.003}$ | $0.309_{0.003}$ | $0.309_{0.004}$ | $0.328_{0.004}$ | $0.355_{0.004}$ |
| | **AUC** | $0.850_{0.001}$ | $0.841_{0.002}$ | $0.835_{0.002}$ | $0.813_{0.002}$ | $0.832_{0.001}$ | $0.834_{0.001}$ | $0.847_{0.002}$ | $0.851_{0.002}$ | $0.863_{0.002}$ | $0.876_{0.002}$ |
| | **AVE** | 3.667 | 5.000 | 6.000 | 8.500 | 6.833 | 7.000 | 3.667 | 2.833 | 1.833 | 1.000 |
| **IAP** | **1-HL** | $0.981_{0.000}$ | $0.981_{0.000}$ | $0.981_{0.000}$ | $0.969_{0.000}$ | $0.980_{0.000}$ | $0.981_{0.000}$ | $0.981_{0.000}$ | $0.981_{0.000}$ | $0.981_{0.000}$ | $0.981_{0.000}$ |
| | **1-OE** | $0.457_{0.007}$ | $0.464_{0.008}$ | $0.454_{0.006}$ | $0.351_{0.008}$ | $0.433_{0.009}$ | $0.377_{0.007}$ | $0.479_{0.007}$ | $0.459_{0.005}$ | $0.491_{0.008}$ | $0.522_{0.009}$ |
| | **1-Cov** | $0.675_{0.004}$ | $0.649_{0.005}$ | $0.630_{0.005}$ | $0.565_{0.004}$ | $0.646_{0.004}$ | $0.605_{0.004}$ | $0.670_{0.004}$ | $0.678_{0.003}$ | $0.706_{0.005}$ | $0.731_{0.004}$ |
| | **1-RL** | $0.884_{0.001}$ | $0.874_{0.002}$ | $0.868_{0.002}$ | $0.833_{0.002}$ | $0.868_{0.002}$ | $0.857_{0.002}$ | $0.882_{0.002}$ | $0.886_{0.001}$ | $0.899_{0.002}$ | $0.910_{0.001}$ |
| | **AP** | $0.329_{0.003}$ | $0.326_{0.003}$ | $0.318_{0.002}$ | $0.236_{0.002}$ | $0.313_{0.004}$ | $0.262_{0.002}$ | $0.340_{0.002}$ | $0.331_{0.003}$ | $0.358_{0.004}$ | $0.387_{0.004}$ |
| | **AUC** | $0.885_{0.001}$ | $0.876_{0.002}$ | $0.870_{0.001}$ | $0.835_{0.001}$ | $0.870_{0.002}$ | $0.859_{0.001}$ | $0.883_{0.002}$ | $0.887_{0.001}$ | $0.899_{0.002}$ | $0.911_{0.001}$ |
| | **AVE** | 4.000 | 4.833 | 6.167 | 8.500 | 6.500 | 7.333 | 3.667 | 3.167 | 1.833 | 1.000 |
| **MIR** | **1-HL** | $0.890_{0.001}$ | $0.890_{0.001}$ | $0.890_{0.001}$ | $0.840_{0.004}$ | $0.880_{0.004}$ | $0.877_{0.001}$ | $0.893_{0.001}$ | $0.890_{0.001}$ | $0.896_{0.001}$ | $0.897_{0.001}$ |
| | **1-OE** | $0.646_{0.009}$ | $0.647_{0.010}$ | $0.645_{0.008}$ | $0.511_{0.016}$ | $0.639_{0.009}$ | $0.609_{0.007}$ | $0.667_{0.006}$ | $0.654_{0.007}$ | $0.690_{0.009}$ | $0.727_{0.003}$ |
| | **1-Cov** | $0.673_{0.003}$ | $0.661_{0.004}$ | $0.657_{0.003}$ | $0.588_{0.013}$ | $0.665_{0.002}$ | $0.624_{0.002}$ | $0.681_{0.002}$ | $0.668_{0.006}$ | $0.694_{0.003}$ | $0.699_{0.003}$ |
| | **1-RL** | $0.874_{0.002}$ | $0.869_{0.003}$ | $0.867_{0.003}$ | $0.809_{0.014}$ | $0.862_{0.003}$ | $0.847_{0.001}$ | $0.878_{0.001}$ | $0.873_{0.002}$ | $0.888_{0.002}$ | $0.897_{0.002}$ |
| | **AP** | $0.599_{0.003}$ | $0.595_{0.007}$ | $0.592_{0.006}$ | $0.494_{0.017}$ | $0.589_{0.004}$ | $0.548_{0.003}$ | $0.614_{0.004}$ | $0.603_{0.005}$ | $0.631_{0.006}$ | $0.653_{0.006}$ |
| | **AUC** | $0.861_{0.001}$ | $0.855_{0.002}$ | $0.854_{0.002}$ | $0.801_{0.017}$ | $0.852_{0.003}$ | $0.839_{0.001}$ | $0.864_{0.001}$ | $0.859_{0.002}$ | $0.875_{0.001}$ | $0.880_{0.001}$ |
| | **AVE** | 4.500 | 5.667 | 6.667 | 9.500 | 7.167 | 8.500 | 3.000 | 4.500 | 2.000 | 1.000 |
| **OBJ** | **1-HL** | $0.948_{0.001}$ | $0.948_{0.001}$ | $0.947_{0.001}$ | $0.899_{0.002}$ | $0.940_{0.003}$ | $0.935_{0.001}$ | $0.949_{0.001}$ | $0.948_{0.001}$ | $0.953_{0.001}$ | $0.951_{0.001}$ |
| | **1-OE** | $0.619_{0.015}$ | $0.601_{0.011}$ | $0.594_{0.012}$ | $0.465_{0.018}$ | $0.604_{0.016}$ | $0.450_{0.008}$ | $0.627_{0.011}$ | $0.626_{0.009}$ | $0.685_{0.011}$ | $0.704_{0.014}$ |
| | **1-Cov** | $0.806_{0.006}$ | $0.794_{0.006}$ | $0.793_{0.006}$ | $0.744_{0.008}$ | $0.796_{0.008}$ | $0.759_{0.006}$ | $0.812_{0.006}$ | $0.809_{0.006}$ | $0.834_{0.007}$ | $0.832_{0.007}$ |
| | **1-RL** | $0.888_{0.005}$ | $0.876_{0.004}$ | $0.875_{0.004}$ | $0.833_{0.006}$ | $0.878_{0.006}$ | $0.850_{0.004}$ | $0.890_{0.005}$ | $0.889_{0.004}$ | $0.910_{0.004}$ | $0.909_{0.004}$ |
| | **AP** | $0.639_{0.010}$ | $0.627_{0.009}$ | $0.623_{0.010}$ | $0.512_{0.014}$ | $0.630_{0.012}$ | $0.537_{0.008}$ | $0.649_{0.009}$ | $0.649_{0.009}$ | $0.692_{0.009}$ | $0.705_{0.010}$ |
| | **AUC** | $0.897_{0.004}$ | $0.886_{0.004}$ | $0.885_{0.004}$ | $0.846_{0.006}$ | $0.888_{0.006}$ | $0.864_{0.004}$ | $0.900_{0.005}$ | $0.898_{0.004}$ | $0.918_{0.004}$ | $0.916_{0.004}$ |
| | **AVE** | 4.667 | 6.333 | 7.333 | 9.333 | 5.833 | 8.667 | 3.000 | 3.833 | 1.333 | 1.667 |
| **PAS** | **1-HL** | $0.931_{0.001}$ | $0.931_{0.000}$ | $0.931_{0.001}$ | $0.882_{0.004}$ | $0.915_{0.005}$ | $0.928_{0.001}$ | $0.933_{0.001}$ | $0.932_{0.001}$ | $0.933_{0.001}$ | $0.935_{0.001}$ |
| | **1-OE** | $0.462_{0.009}$ | $0.443_{0.007}$ | $0.435_{0.010}$ | $0.366_{0.039}$ | $0.433_{0.017}$ | $0.418_{0.008}$ | $0.473_{0.008}$ | $0.468_{0.008}$ | $0.495_{0.013}$ | $0.526_{0.013}$ |
| | **1-Cov** | $0.781_{0.007}$ | $0.749_{0.003}$ | $0.738_{0.010}$ | $0.674_{0.011}$ | $0.759_{0.006}$ | $0.738_{0.003}$ | $0.790_{0.006}$ | $0.778_{0.004}$ | $0.817_{0.004}$ | $0.820_{0.006}$ |
| | **1-RL** | $0.830_{0.006}$ | $0.803_{0.002}$ | $0.792_{0.008}$ | $0.736_{0.011}$ | $0.808_{0.006}$ | $0.797_{0.002}$ | $0.836_{0.005}$ | $0.828_{0.004}$ | $0.862_{0.004}$ | $0.867_{0.005}$ |
| | **AP** | $0.548_{0.007}$ | $0.517_{0.004}$ | $0.510_{0.008}$ | $0.438_{0.022}$ | $0.524_{0.009}$ | $0.486_{0.005}$ | $0.562_{0.005}$ | $0.552_{0.006}$ | $0.590_{0.008}$ | $0.607_{0.008}$ |
| | **AUC** | $0.851_{0.005}$ | $0.827_{0.002}$ | $0.817_{0.008}$ | $0.767_{0.011}$ | $0.830_{0.006}$ | $0.823_{0.002}$ | $0.855_{0.005}$ | $0.848_{0.005}$ | $0.880_{0.003}$ | $0.883_{0.004}$ |
| | **AVE** | 4.333 | 6.333 | 7.500 | 9.333 | 6.333 | 7.833 | 2.833 | 4.333 | 2.000 | 1.000 |

set to $\{30\%, 50\%, 70\%, 90\%\}$. Table 1 reports the mean and standard deviation at 50% PER and 50% LMR, along with the average ranking across all metrics. Fig. 2 illustrates AP variations as PER and LMR change from 30% to 90%.

Drawing from the comparison results, we have the following observations: (i) Our method exhibits outstanding performance on almost all metrics across all six datasets. As shown in Table 1, despite the fluctuating rankings of other methods, LGRL holds the top position on five of the six datasets and ranks second on OBJECT, demonstrating its effectiveness in addressing the iMvMLC problem. (ii) SIP, MTD, and TDLSR are competitive methods that frequently appear among the top three. Our method surpasses these approaches by incorporating category-specific priors into

Bayesian fusion, which enables hypothesis-based reasoning tailored to each label. On Corel5k, LGRL achieves an AP of 0.469, outperforming the second-best method TDLSR (0.450) by 4.2%, and surpassing MTD and SIP (0.414) by 13.3%. Similar improvements over TDLSR are observed on ESPGame and IAPRTC12, with AP gains of 8.2% and 8.1%, respectively. (iii) As depicted in Fig. 2, our method exhibits strong robustness across a wide range of missing ratios and is particularly well-suited for highly incomplete settings. For instance, when PER reaches 90% on Corel5k and ESPGame, most baseline methods suffer severe degradation, while LGRL continues to deliver stable results. A detailed comparison under this extreme 90% PER and LMR setting is further provided in Table 4.

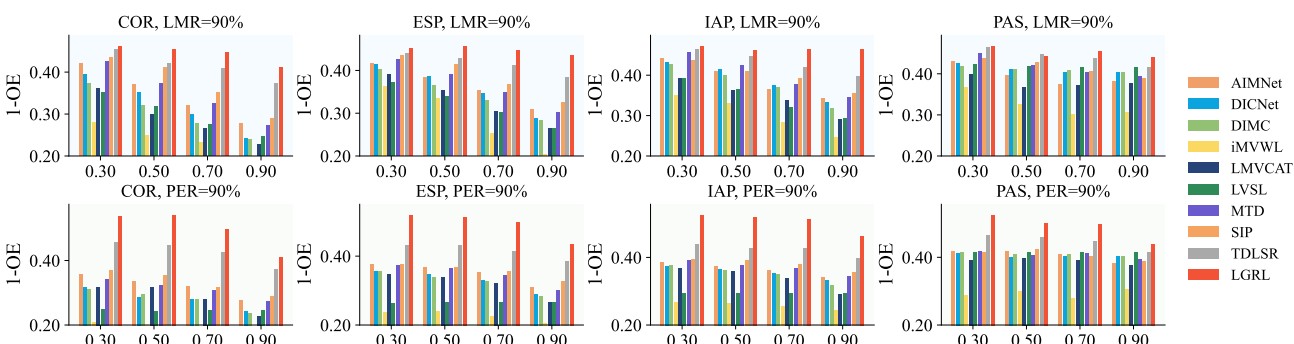

*Figure 2.* Experimental results on four datasets with PER and LMR changing from 30% to 90%.

*Table 2.* Ablation study on four datasets with PER=50% and LMR=50%. '✓' and '✗' indicate whether the corresponding module is used.

| $S_1$ | $S_2$ | $S_3$ | Corel5k | | | | Pascal07 | | | | OBJECT | | | | Mirflickr | | | |
|---|---|---|---|---|---|---|---|---|---|---|---|---|---|---|---|---|---|---|
| | | | AP | AUC | 1-RL | 1-OE | AP | AUC | 1-RL | 1-OE | AP | AUC | 1-RL | 1-OE | AP | AUC | 1-RL | 1-OE |
| ✗ | ✓ | ✓ | 0.464 | 0.922 | 0.920 | 0.560 | 0.586 | 0.869 | 0.853 | 0.502 | 0.680 | 0.904 | 0.897 | 0.678 | 0.638 | 0.871 | 0.888 | 0.710 |
| ✓ | ✗ | ✓ | 0.455 | 0.916 | 0.914 | 0.548 | 0.582 | 0.866 | 0.850 | 0.498 | 0.676 | 0.901 | 0.894 | 0.672 | 0.633 | 0.867 | 0.884 | 0.703 |
| ✓ | ✓ | ✗ | 0.446 | 0.912 | 0.910 | 0.538 | 0.548 | 0.842 | 0.825 | 0.462 | 0.635 | 0.876 | 0.868 | 0.632 | 0.608 | 0.849 | 0.865 | 0.678 |
| ✓ | ✓ | ✓ | **0.469** | **0.925** | **0.923** | **0.567** | **0.607** | **0.883** | **0.867** | **0.526** | **0.705** | **0.916** | **0.909** | **0.704** | **0.653** | **0.880** | **0.897** | **0.727** |

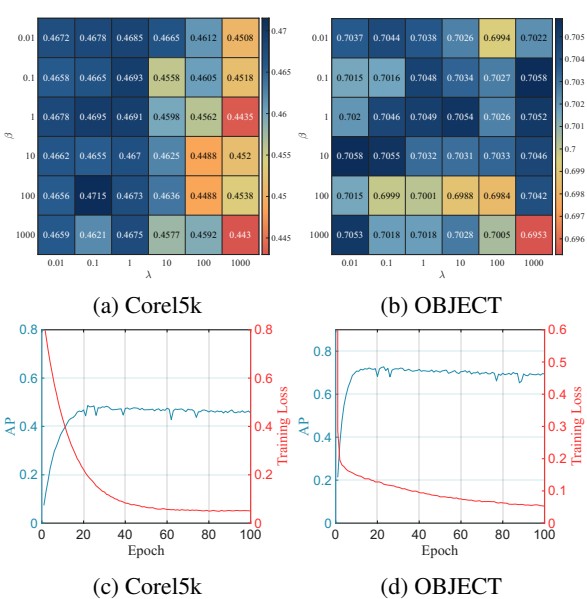

(a) Corel5k

(b) OBJECT

(c) Corel5k

(d) OBJECT

*Figure 3.* Parameter sensitivity of $\lambda$ and $\beta$ (a–b) and convergence behavior during training (c–d).

### 3.3. Ablation Study

To evaluate the contribution of each component in LGRL, we conduct ablation experiments by systematically removing the semantic alignment loss ($S_1$), task-relevant consistency constraints ($S_2$), and category-specific conditional posteriors ($S_3$). Specifically, removing $S_1$ discards $\mathcal{L}_{SA}$ from the training objective; removing $S_2$ eliminates both encoder and classifier consistency terms; and removing $S_3$ excludes category prototypes from Bayesian fusion, reducing

it to standard PoE over view posteriors without label-guided priors. As shown in Table 2, the full model consistently achieves the best performance, validating the necessity of all three components. Among them, $S_3$ has the most pronounced impact, confirming that incorporating category-specific priors into fusion enables more effective hypothesis-based reasoning. The consistency constraints $S_2$ contributes consistent improvements across datasets. The semantic alignment loss $S_1$ provides considerable improvements on Pascal07 and OBJECT, though its effect on Corel5k is less significant, likely due to the increased difficulty of aligning representations across a larger label space.

### 3.4. Parameter Sensitivity and Convergence

We analyze the sensitivity of LGRL to hyperparameters $\lambda$ and $\beta$, which balance feature-level and label-level objectives, and control semantic alignment strength, respectively. As shown in Fig. 3(a–b), LGRL achieves stable performance when both parameters fall within $(0.1, 1)$, demonstrating robustness to hyperparameter selection. Fig. 3(c–d) illustrates the convergence behavior, showing that LGRL converges rapidly within the first 20 epochs and maintains stable performance thereafter.

## 4. Applications

### 4.1. NBA Player Potential Prediction

Evaluating player potential requires integrating heterogeneous performance statistics while predicting multiple attributes simultaneously, including career trajectory, positional role, and award likelihood. In practice, data collec-

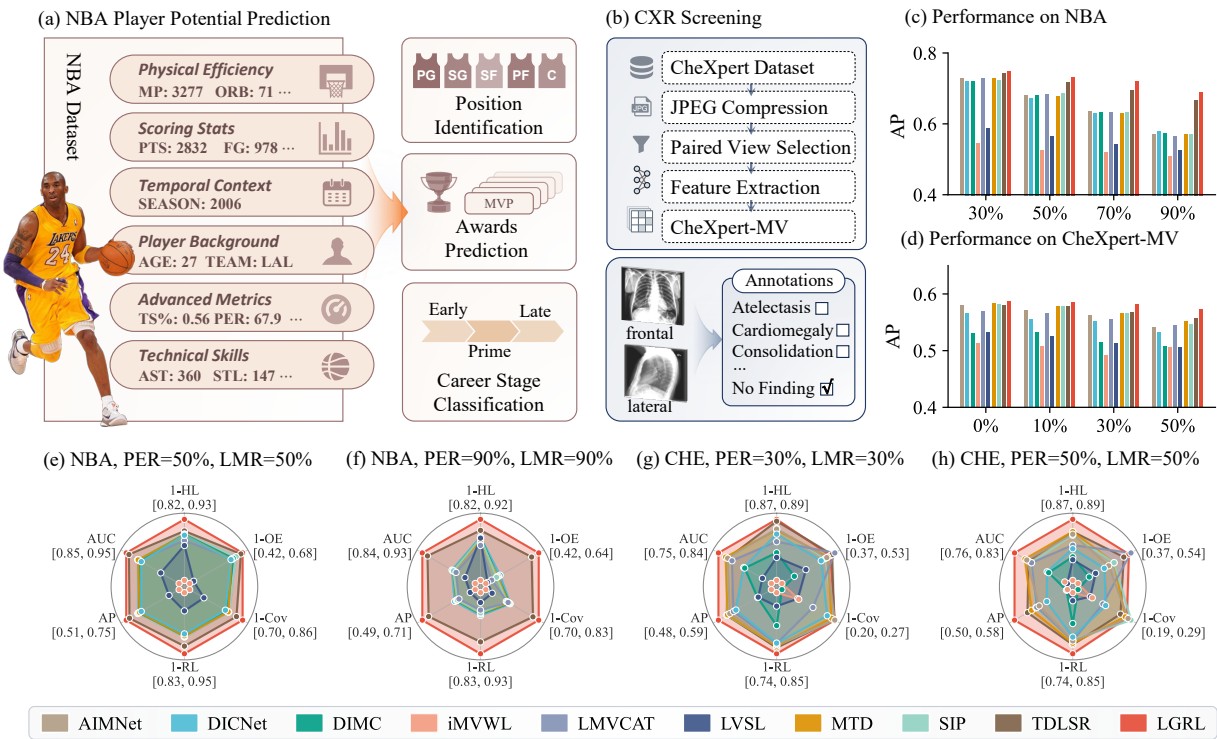

*Figure 4.* Overview of real-world applications and experimental performance. (a)–(b) Workflows for NBA player potential prediction and chest radiograph screening. (c)–(d) Performance (AP) across varying missing ratios. X-axis represents synchronized Partial Example Ratio (PER) and Label Missing Ratio (LMR) (e.g., 30% denotes PER=LMR=30%). (e)–(h) Comprehensive metric comparisons using radar charts under specific missingness settings.

tion across a player's career is inevitably incomplete due to injuries, trades, and evolving statistical standards.

We construct an NBA player dataset from Basketball-Reference[1], comprising 16,992 player-season records spanning 2002–2022. Each sample aggregates six statistical views: scoring, rebounding and physical metrics, playmaking, advanced efficiency metrics, player background, and season context. The prediction targets include career stage classification (early, prime, or late phase based on professional timeline tertiles), positional identification (PG, SG, SF, PF, C), and award predictions (MVP, Defensive Player of the Year, etc.).

We evaluate all methods under PER and LMR ranging from 30% to 90%. As visualized in Figure 4(c), LGRL maintains a distinct performance margin in Average Precision across this wide range of missingness. Furthermore, the radar charts in Figure 4(e)–(f) demonstrate that our method achieves comprehensive improvements across six evaluation metrics (e.g., AUC, Ranking Loss), exhibiting strong robustness under both moderate (50%) and severe (90%) missingness settings.

---

[1] Data available at https://www.basketball-reference.com.

## 4.2. Chest Radiograph Disease Screening

Chest radiographs serve as a crucial frontline tool for preliminary disease screening. Standard chest imaging includes frontal and lateral projections, where the latter can reveal pathologies obscured on frontal views, such as retrocardiac lesions and small pleural effusions. However, lateral views are sometimes difficult to obtain due to patient immobility, bedside portable imaging, or limited resources in primary care settings. Label incompleteness is also particularly common in medical imaging, as comprehensive annotation demands substantial specialist effort.

We derive our dataset from CheXpert (Irvin et al., 2019), selecting 31,413 studies containing both frontal and lateral chest radiographs as two views. The images are converted to JPEG format, and view-specific features are extracted using a pre-trained DenseNet-121 (Huang et al., 2017) with the classification layer removed, yielding 1,024-dimensional representations per view. The label space comprises 14 pathological observations including cardiomegaly, pneumonia, pleural effusion, and pneumothorax. Both view and label missingness are simulated at varying ratios to accurately reflect realistic clinical scenarios.

Figure 4(d) illustrates that LGRL consistently achieves the highest AP scores across synchronous missing ratios from 0% to 50%. Additionally, the radar charts in Figure 4(g)–(h) confirm that LGRL yields a balanced performance profile with significant gains in ranking and coverage metrics, validating its effectiveness in identifying pathologies despite incomplete views and labels. The improvements achieved by LGRL suggest its potential for automated preliminary screening, providing clinical prompts that assist radiologists in identifying findings that might otherwise be overlooked under incomplete imaging protocols.

## 5. Conclusion

In this paper, we propose Label-Guided Representation Learning (LGRL) for incomplete multi-view multi-label classification. LGRL systematically exploits label semantics throughout the learning pipeline by introducing learnable category prototypes that impose geometric structure on the latent space via contrastive alignment, formulating task-relevant objectives that ensure both reconstruction and classification while enforcing cross-view consistency, and designing category-specific conditional posteriors for hypothesis-based multi-label reasoning. Extensive experiments on benchmark datasets and real-world applications demonstrate the superiority of LGRL, though prototype learning currently scales linearly with $C$ and prototype clustering would be a natural extension for extremely large label spaces. We plan to explore richer semantic sources to enhance label semantic learning and prototype initialization for iMvMLC.

## Acknowledgements

This work was partially supported by the National Natural Science Foundation of China Grant No. 62376281.

## Impact Statement

This paper aims to advance machine learning for incomplete multi-view multi-label data. We demonstrate its utility in medical imaging (e.g., chest radiograph screening). While beneficial for handling missing modalities, practitioners should carefully evaluate model reliability and potential biases before deploying in high-stakes clinical settings.

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

# A. Derivation of Variational Bounds

This appendix provides rigorous derivations for the variational bounds employed in the task-relevant representation learning objective (Section 2.3). The theoretical foundation rests on the information bottleneck principle, which seeks to balance two competing objectives: *sufficiency*, the retention of task-relevant information, and *consistency*, the suppression of view-specific redundancy.

## A.1. Problem Formulation

Let $\mathbf{x}^{(v)} \in \mathbb{R}^{d_v}$ denote the input from view $v$, $\mathbf{z}^{(v)} \in \mathbb{R}^{d_z}$ the corresponding view-specific latent representation, $\mathbf{z}$ the fused representation aggregating all available views, $\mathbf{x}^{\sim(v)}$ all views excluding view $v$, $\mathbf{z}^{\sim(v)}$ the latent representations from all views excluding view $v$, and $\mathbf{y} \in \{0, 1\}^C$ the multi-label annotation. The variational encoder is denoted $q_\phi(\mathbf{z}^{(v)} \mid \mathbf{x}^{(v)})$, the variational decoder $p_\theta(\mathbf{x}^{(v)} \mid \mathbf{z}^{(v)})$, the fused posterior $q(\mathbf{z} \mid \{\mathbf{x}\})$, and the variational classifier $p_\psi(\mathbf{y} \mid \mathbf{z})$.

Following the information bottleneck principle, our objective is to learn representations that maximally preserve task-relevant information while discarding view-specific redundancy. This is formalized through the following optimization problem:

$$\max_{\phi,\theta,\psi} \sum_{v \in \mathcal{V}} \left[ \mathcal{I}_{\text{feat}}^{(v)} + \lambda \cdot \mathcal{I}_{\text{label}}^{(v)} \right], \tag{19}$$

where the feature-level and label-level information objectives are defined as:

$$\mathcal{I}_{\text{feat}}^{(v)} = \underbrace{I(\mathbf{x}^{(v)}; \mathbf{z}^{(v)})}_{\text{Feature Sufficiency}} - \alpha \underbrace{I(\mathbf{x}^{\sim(v)}; \mathbf{z}^{(v)} \mid \mathbf{x}^{(v)})}_{\text{Feature Consistency}}, \tag{20}$$

$$\mathcal{I}_{\text{label}}^{(v)} = \underbrace{I(\mathbf{y}; \mathbf{z}^{(v)})}_{\text{Label Sufficiency}} - \gamma \underbrace{I(\mathbf{y}; \mathbf{z}^{(v)} \mid \mathbf{z}^{\sim(v)})}_{\text{Label Consistency}}. \tag{21}$$

The objective decomposes into four mutual information terms:

- **Feature Sufficiency** $I(\mathbf{x}^{(v)}; \mathbf{z}^{(v)})$: Ensures faithful reconstruction of the input.

- **Feature Consistency** $I(\mathbf{x}^{\sim(v)}; \mathbf{z}^{(v)} \mid \mathbf{x}^{(v)})$: Enforces cross-view agreement at the representation level.

- **Label Sufficiency** $I(\mathbf{y}; \mathbf{z}^{(v)})$: Preserves discriminative information for classification.

- **Label Consistency** $I(\mathbf{y}; \mathbf{z}^{(v)} \mid \mathbf{z}^{\sim(v)})$: Enforces semantic agreement across views.

Direct optimization of these mutual information terms is intractable. We employ variational inference to derive tractable bounds: for sufficiency terms (to be maximized), we derive variational *lower bounds*; for consistency terms (to be minimized), we derive variational *upper bounds*.

## A.2. Feature Sufficiency

We seek to maximize the mutual information $I(\mathbf{x}^{(v)}; \mathbf{z}^{(v)})$ between the observed view and its latent encoding.

**Proposition A.1.** *The mutual information $I(\mathbf{x}^{(v)}; \mathbf{z}^{(v)})$ admits the following variational lower bound:*

$$I(\mathbf{x}^{(v)}; \mathbf{z}^{(v)}) \geq \mathbb{E}_{p(\mathbf{x}^{(v)})} \mathbb{E}_{q_\phi(\mathbf{z}^{(v)} \mid \mathbf{x}^{(v)})} \left[ \log p_\theta(\mathbf{x}^{(v)} \mid \mathbf{z}^{(v)}) \right] + H(\mathbf{x}^{(v)}), \tag{22}$$

*where $H(\mathbf{x}^{(v)})$ denotes the differential entropy of the data distribution.*

*Proof.* By the definition of mutual information:

$$I(\mathbf{x}^{(v)}; \mathbf{z}^{(v)}) = \mathbb{E}_{p(\mathbf{x}^{(v)}, \mathbf{z}^{(v)})} \left[ \log \frac{p(\mathbf{x}^{(v)}, \mathbf{z}^{(v)})}{p(\mathbf{x}^{(v)}) p(\mathbf{z}^{(v)})} \right]. \tag{23}$$

Applying Bayes' rule to the joint distribution $p(\mathbf{x}^{(v)}, \mathbf{z}^{(v)}) = p(\mathbf{x}^{(v)} \mid \mathbf{z}^{(v)})p(\mathbf{z}^{(v)})$:

$$
\begin{aligned}
I(\mathbf{x}^{(v)}; \mathbf{z}^{(v)}) &= \mathbb{E}_{p(\mathbf{x}^{(v)}, \mathbf{z}^{(v)})} \left[ \log \frac{p(\mathbf{x}^{(v)} \mid \mathbf{z}^{(v)})}{p(\mathbf{x}^{(v)})} \right] \\
&= \mathbb{E}_{p(\mathbf{x}^{(v)}, \mathbf{z}^{(v)})} \left[ \log p(\mathbf{x}^{(v)} \mid \mathbf{z}^{(v)}) \right] - \mathbb{E}_{p(\mathbf{x}^{(v)})} \left[ \log p(\mathbf{x}^{(v)}) \right] \\
&= \mathbb{E}_{p(\mathbf{x}^{(v)}, \mathbf{z}^{(v)})} \left[ \log p(\mathbf{x}^{(v)} \mid \mathbf{z}^{(v)}) \right] + H(\mathbf{x}^{(v)}).
\end{aligned}
\tag{24}
$$

The entropy term $H(\mathbf{x}^{(v)})$ is constant with respect to model parameters. We focus on maximizing:

$$
\mathcal{J}_{\text{recon}} = \mathbb{E}_{p(\mathbf{x}^{(v)}, \mathbf{z}^{(v)})} \left[ \log p(\mathbf{x}^{(v)} \mid \mathbf{z}^{(v)}) \right].
\tag{25}
$$

The true conditional likelihood $p(\mathbf{x}^{(v)} \mid \mathbf{z}^{(v)})$ is intractable. Introducing a variational decoder $p_\theta(\mathbf{x}^{(v)} \mid \mathbf{z}^{(v)})$ and applying the non-negativity of KL divergence:

$$
\mathbb{E}_{p(\mathbf{x}^{(v)}, \mathbf{z}^{(v)})} \left[ \log p(\mathbf{x}^{(v)} \mid \mathbf{z}^{(v)}) \right] \geq \mathbb{E}_{p(\mathbf{x}^{(v)}, \mathbf{z}^{(v)})} \left[ \log p_\theta(\mathbf{x}^{(v)} \mid \mathbf{z}^{(v)}) \right].
\tag{26}
$$

Replacing the true posterior $p(\mathbf{z}^{(v)} \mid \mathbf{x}^{(v)})$ with the variational encoder $q_\phi(\mathbf{z}^{(v)} \mid \mathbf{x}^{(v)})$:

$$
\mathcal{J}_{\text{recon}} \geq \mathbb{E}_{p(\mathbf{x}^{(v)})} \mathbb{E}_{q_\phi(\mathbf{z}^{(v)} \mid \mathbf{x}^{(v)})} \left[ \log p_\theta(\mathbf{x}^{(v)} \mid \mathbf{z}^{(v)}) \right].
\tag{27}
$$

$\square$

This establishes the lower bound in Eq. (6). The corresponding **Reconstruction Loss** is:

$$
\mathcal{L}_{\text{recon}}^{(v)} = -\mathbb{E}_{q_\phi(\mathbf{z}^{(v)} \mid \mathbf{x}^{(v)})} \left[ \log p_\theta(\mathbf{x}^{(v)} \mid \mathbf{z}^{(v)}) \right].
\tag{28}
$$

### A.3. Feature Consistency

We try to obtain feature consistency by minimizing $I(\mathbf{x}^{\sim(v)}; \mathbf{z}^{(v)} \mid \mathbf{x}^{(v)})$, which quantifies the information that $\mathbf{z}^{(v)}$ retains about other views beyond what is captured by $\mathbf{x}^{(v)}$.

**Proposition A.2.** *The conditional mutual information $I(\mathbf{x}^{\sim(v)}; \mathbf{z}^{(v)} \mid \mathbf{x}^{(v)})$ is upper-bounded as:*

$$
I(\mathbf{x}^{\sim(v)}; \mathbf{z}^{(v)} \mid \mathbf{x}^{(v)}) \leq \mathbb{E}_{p(\{\mathbf{x}\})} \left[ D_{\text{KL}} \left( q(\mathbf{z} \mid \{\mathbf{x}\}) \,\|\, q_\phi(\mathbf{z} \mid \mathbf{x}^{(v)}) \right) \right].
\tag{29}
$$

*Proof.* We first establish a relaxation. Since the fused representation $\mathbf{z}$ aggregates information from all views, it contains at least as much information about $\mathbf{x}^{\sim(v)}$ as $\mathbf{z}^{(v)}$ does. By the data processing inequality:

$$
I(\mathbf{x}^{\sim(v)}; \mathbf{z}^{(v)} \mid \mathbf{x}^{(v)}) \leq I(\mathbf{x}^{\sim(v)}; \mathbf{z} \mid \mathbf{x}^{(v)}).
\tag{30}
$$

By the definition of conditional mutual information:

$$
I(\mathbf{x}^{\sim(v)}; \mathbf{z} \mid \mathbf{x}^{(v)}) = \mathbb{E}_{p(\{\mathbf{x}\}, \mathbf{z})} \left[ \log \frac{p(\mathbf{x}^{\sim(v)}, \mathbf{z} \mid \mathbf{x}^{(v)})}{p(\mathbf{x}^{\sim(v)} \mid \mathbf{x}^{(v)})p(\mathbf{z} \mid \mathbf{x}^{(v)})} \right].
\tag{31}
$$

Applying Bayes' rule: $p(\mathbf{x}^{\sim(v)}, \mathbf{z} \mid \mathbf{x}^{(v)}) = p(\mathbf{z} \mid \{\mathbf{x}\})p(\mathbf{x}^{\sim(v)} \mid \mathbf{x}^{(v)})$. Substituting and simplifying:

$$
\begin{aligned}
I(\mathbf{x}^{\sim(v)}; \mathbf{z} \mid \mathbf{x}^{(v)}) &= \mathbb{E}_{p(\{\mathbf{x}\}, \mathbf{z})} \left[ \log \frac{p(\mathbf{z} \mid \{\mathbf{x}\})}{p(\mathbf{z} \mid \mathbf{x}^{(v)})} \right] \\
&= \mathbb{E}_{p(\{\mathbf{x}\})} \left[ D_{\text{KL}} \left( p(\mathbf{z} \mid \{\mathbf{x}\}) \,\|\, p(\mathbf{z} \mid \mathbf{x}^{(v)}) \right) \right].
\end{aligned}
\tag{32}
$$

The true posteriors are intractable. Introducing variational approximations $q(\mathbf{z} \mid \{\mathbf{x}\})$ and $q_\phi(\mathbf{z} \mid \mathbf{x}^{(v)})$, we decompose:

$$
\begin{aligned}
& D_{\mathrm{KL}} \left( p(\mathbf{z} \mid \{\mathbf{x}\}) \,\|\, p(\mathbf{z} \mid \mathbf{x}^{(v)}) \right) \\
&= \mathbb{E}_{p(\mathbf{z}|\{\mathbf{x}\})} \left[ \log \frac{p(\mathbf{z} \mid \{\mathbf{x}\})}{q_\phi(\mathbf{z} \mid \mathbf{x}^{(v)})} + \log \frac{q_\phi(\mathbf{z} \mid \mathbf{x}^{(v)})}{p(\mathbf{z} \mid \mathbf{x}^{(v)})} \right] \\
&= D_{\mathrm{KL}} \left( p(\mathbf{z} \mid \{\mathbf{x}\}) \,\|\, q_\phi(\mathbf{z} \mid \mathbf{x}^{(v)}) \right) - D_{\mathrm{KL}} \left( p(\mathbf{z} \mid \mathbf{x}^{(v)}) \,\|\, q_\phi(\mathbf{z} \mid \mathbf{x}^{(v)}) \right).
\end{aligned}
\tag{33}
$$

Note that the second term involves expectation under $p(\mathbf{z} \mid \{\mathbf{x}\})$ while the KL divergence is defined under $p(\mathbf{z} \mid \mathbf{x}^{(v)})$. For analytical tractability, we invoke the assumption that the view-specific encoder provides a sufficiently accurate approximation. Dropping the non-positive second term yields:

$$
D_{\mathrm{KL}} \left( p(\mathbf{z} \mid \{\mathbf{x}\}) \,\|\, p(\mathbf{z} \mid \mathbf{x}^{(v)}) \right) \leq D_{\mathrm{KL}} \left( p(\mathbf{z} \mid \{\mathbf{x}\}) \,\|\, q_\phi(\mathbf{z} \mid \mathbf{x}^{(v)}) \right).
\tag{34}
$$

Replacing the true fused posterior with its variational approximation $q(\mathbf{z} \mid \{\mathbf{x}\})$:

$$
I(\mathbf{x}^{\sim(v)}; \mathbf{z}^{(v)} \mid \mathbf{x}^{(v)}) \leq \mathbb{E}_{p(\{\mathbf{x}\})} \left[ D_{\mathrm{KL}} \left( q(\mathbf{z} \mid \{\mathbf{x}\}) \,\|\, q_\phi(\mathbf{z} \mid \mathbf{x}^{(v)}) \right) \right].
\tag{35}
$$

$\square$

This establishes the upper bound in Eq. (7). The corresponding **Encoder Consistency Loss** is:

$$
\mathcal{L}_{\text{enc-consist}}^{(v)} = D_{\mathrm{KL}} \left( q(\mathbf{z} \mid \{\mathbf{x}\}) \,\|\, q_\phi(\mathbf{z} \mid \mathbf{x}^{(v)}) \right).
\tag{36}
$$

For Gaussian distributions with diagonal covariance, $q(\mathbf{z} \mid \{\mathbf{x}\}) = \mathcal{N}(\boldsymbol{\mu}_{\text{fused}}, \operatorname{diag}(\boldsymbol{\sigma}_{\text{fused}}^2))$ and $q_\phi(\mathbf{z} \mid \mathbf{x}^{(v)}) = \mathcal{N}(\boldsymbol{\mu}_v, \operatorname{diag}(\boldsymbol{\sigma}_v^2))$, this KL divergence admits a closed-form expression:

$$
D_{\mathrm{KL}} = \frac{1}{2} \sum_{j=1}^{d_z} \left[ \log \frac{\sigma_{v,j}^2}{\sigma_{\text{fused},j}^2} + \frac{\sigma_{\text{fused},j}^2 + (\mu_{\text{fused},j} - \mu_{v,j})^2}{\sigma_{v,j}^2} - 1 \right].
\tag{37}
$$

### A.4. Label Sufficiency

We seek to maximize $I(\mathbf{y}; \mathbf{z}^{(v)})$, the mutual information between labels and the view-specific latent representation.

**Proposition A.3.** *The mutual information $I(\mathbf{y}; \mathbf{z}^{(v)})$ admits the following variational lower bound:*

$$
I(\mathbf{y}; \mathbf{z}^{(v)}) \geq \mathbb{E}_{p(\mathbf{x}^{(v)})} \mathbb{E}_{q_\phi(\mathbf{z}^{(v)}|\mathbf{x}^{(v)})} \mathbb{E}_{p(\mathbf{y}|\mathbf{x}^{(v)})} \left[ \log p_\psi(\mathbf{y} \mid \mathbf{z}^{(v)}) \right] + H(\mathbf{y}).
\tag{38}
$$

*Proof.* By the definition of mutual information:

$$
\begin{aligned}
I(\mathbf{y}; \mathbf{z}^{(v)}) &= \mathbb{E}_{p(\mathbf{y}, \mathbf{z}^{(v)})} \left[ \log \frac{p(\mathbf{y} \mid \mathbf{z}^{(v)})}{p(\mathbf{y})} \right] \\
&= \mathbb{E}_{p(\mathbf{y}, \mathbf{z}^{(v)})} \left[ \log p(\mathbf{y} \mid \mathbf{z}^{(v)}) \right] + H(\mathbf{y}).
\end{aligned}
\tag{39}
$$

The entropy $H(\mathbf{y})$ is independent of model parameters. We focus on maximizing:

$$
\mathcal{J}_{\text{cls}} = \mathbb{E}_{p(\mathbf{y}, \mathbf{z}^{(v)})} \left[ \log p(\mathbf{y} \mid \mathbf{z}^{(v)}) \right].
\tag{40}
$$

Introducing a variational classifier $p_\psi(\mathbf{y} \mid \mathbf{z}^{(v)})$ and applying the non-negativity of KL divergence:

$$
\mathcal{J}_{\text{cls}} \geq \mathbb{E}_{p(\mathbf{y}, \mathbf{z}^{(v)})} \left[ \log p_\psi(\mathbf{y} \mid \mathbf{z}^{(v)}) \right].
\tag{41}
$$

Expanding over the data distribution by introducing $\mathbf{x}^{(v)}$:

$$\mathbb{E}_{p(\mathbf{y}, \mathbf{z}^{(v)})} \left[ \log p_\psi(\mathbf{y} \mid \mathbf{z}^{(v)}) \right] = \iiint p(\mathbf{y}, \mathbf{z}^{(v)}, \mathbf{x}^{(v)}) \log p_\psi(\mathbf{y} \mid \mathbf{z}^{(v)}) \, d\mathbf{y} \, d\mathbf{z}^{(v)} \, d\mathbf{x}^{(v)}. \tag{42}$$

Under the conditional independence assumption $p(\mathbf{y}, \mathbf{z}^{(v)} \mid \mathbf{x}^{(v)}) = p(\mathbf{y} \mid \mathbf{x}^{(v)})p(\mathbf{z}^{(v)} \mid \mathbf{x}^{(v)})$:

$$\mathbb{E}_{p(\mathbf{y}, \mathbf{z}^{(v)})} \left[ \log p_\psi(\mathbf{y} \mid \mathbf{z}^{(v)}) \right] = \mathbb{E}_{p(\mathbf{x}^{(v)})} \mathbb{E}_{p(\mathbf{z}^{(v)} \mid \mathbf{x}^{(v)})} \mathbb{E}_{p(\mathbf{y} \mid \mathbf{x}^{(v)})} \left[ \log p_\psi(\mathbf{y} \mid \mathbf{z}^{(v)}) \right]. \tag{43}$$

Replacing the true posterior with the variational encoder:

$$\mathcal{J}_{\text{cls}} \geq \mathbb{E}_{p(\mathbf{x}^{(v)})} \mathbb{E}_{q_\phi(\mathbf{z}^{(v)} \mid \mathbf{x}^{(v)})} \mathbb{E}_{p(\mathbf{y} \mid \mathbf{x}^{(v)})} \left[ \log p_\psi(\mathbf{y} \mid \mathbf{z}^{(v)}) \right]. \tag{44}$$

$$\square$$

This establishes the lower bound in Eq. (8). The corresponding **Classification Loss** is:

$$\mathcal{L}_{\text{cls}}^{(v)} = -\mathbb{E}_{q_\phi(\mathbf{z}^{(v)} \mid \mathbf{x}^{(v)})} \mathbb{E}_{p(\mathbf{y} \mid \mathbf{x}^{(v)})} \left[ \log p_\psi(\mathbf{y} \mid \mathbf{z}^{(v)}) \right]. \tag{45}$$

For multi-label classification with partial labels, assuming conditional independence across labels, the classifier factorizes as $p_\psi(\mathbf{y} \mid \mathbf{z}^{(v)}) = \prod_{c=1}^{C} p_\psi(y_c \mid \mathbf{z}^{(v)})$. In the supervised setting where labels are observed, the inner expectation $\mathbb{E}_{p(\mathbf{y} \mid \mathbf{x}^{(v)})}$ collapses to evaluation at the observed labels $\mathcal{O}_i$, yielding the binary cross-entropy form:

$$\mathcal{L}_{\text{cls}}^{(v)} = -\frac{1}{|\mathcal{O}_i|} \sum_{c \in \mathcal{O}_i} \left[ y_c \log \hat{y}_c^{(v)} + (1 - y_c) \log(1 - \hat{y}_c^{(v)}) \right], \tag{46}$$

where $\hat{y}_c^{(v)} = \sigma(f_c(\mathbf{z}^{(v)}))$ denotes the predicted probability for category $c$.

### A.5. Label Consistency

We seek to minimize $I(\mathbf{y}; \mathbf{z}^{(v)} \mid \mathbf{z}^{\sim(v)})$, which measures the redundant label information in $\mathbf{z}^{(v)}$ given the representations from other views.

**Proposition A.4.** *The conditional mutual information $I(\mathbf{y}; \mathbf{z}^{(v)} \mid \mathbf{z}^{\sim(v)})$ is upper-bounded as:*

$$I(\mathbf{y}; \mathbf{z}^{(v)} \mid \mathbf{z}^{\sim(v)}) \leq \mathbb{E}_{p(\{\mathbf{x}\})} \mathbb{E}_{q(\mathbf{z} \mid \{\mathbf{x}\})} \left[ D_{\text{KL}} \left( p_\psi(\mathbf{y} \mid \mathbf{z}) \,\|\, p_\psi(\mathbf{y} \mid \mathbf{z}^{(v)}) \right) \right]. \tag{47}$$

*Proof.* By the definition of conditional mutual information:

$$I(\mathbf{y}; \mathbf{z}^{(v)} \mid \mathbf{z}^{\sim(v)}) = \mathbb{E}_{p(\mathbf{y}, \mathbf{z}^{(v)}, \mathbf{z}^{\sim(v)})} \left[ \log \frac{p(\mathbf{y}, \mathbf{z}^{(v)} \mid \mathbf{z}^{\sim(v)})}{p(\mathbf{y} \mid \mathbf{z}^{\sim(v)}) p(\mathbf{z}^{(v)} \mid \mathbf{z}^{\sim(v)})} \right]. \tag{48}$$

Applying Bayes' rule: $p(\mathbf{y}, \mathbf{z}^{(v)} \mid \mathbf{z}^{\sim(v)}) = p(\mathbf{y} \mid \mathbf{z}^{(v)}, \mathbf{z}^{\sim(v)}) p(\mathbf{z}^{(v)} \mid \mathbf{z}^{\sim(v)})$. Substituting and simplifying:

$$I(\mathbf{y}; \mathbf{z}^{(v)} \mid \mathbf{z}^{\sim(v)}) = \mathbb{E}_{p(\mathbf{y}, \{\mathbf{z}\})} \left[ \log \frac{p(\mathbf{y} \mid \mathbf{z}^{(v)}, \mathbf{z}^{\sim(v)})}{p(\mathbf{y} \mid \mathbf{z}^{\sim(v)})} \right]. \tag{49}$$

Under the assumption $p(\mathbf{y} \mid \mathbf{z}^{(v)}, \mathbf{z}^{\sim(v)}) = p(\mathbf{y} \mid \{\mathbf{z}\})$:

$$
\begin{aligned}
I(\mathbf{y}; \mathbf{z}^{(v)} \mid \mathbf{z}^{\sim(v)}) &= \mathbb{E}_{p(\{\mathbf{z}\})} \left[ D_{\text{KL}} \left( p(\mathbf{y} \mid \{\mathbf{z}\}) \,\|\, p(\mathbf{y} \mid \mathbf{z}^{\sim(v)}) \right) \right] \\
&= \mathbb{E}_{p(\{\mathbf{x}\})} \mathbb{E}_{p(\mathbf{z} \mid \{\mathbf{x}\})} \left[ D_{\text{KL}} \left( p(\mathbf{y} \mid \{\mathbf{z}\}) \,\|\, p(\mathbf{y} \mid \mathbf{z}^{\sim(v)}) \right) \right].
\end{aligned} \tag{50}
$$

Direct computation over $\mathbf{z}^{\sim(v)}$ is cumbersome. We adopt a symmetric formulation: for each view $v$, we measure the divergence between predictions from the fused representation and predictions from view $v$ alone. The rationale is that if $p(\mathbf{y} \mid \mathbf{z}^{(v)})$ differs significantly from $p(\mathbf{y} \mid \{\mathbf{z}\})$, then $\mathbf{z}^{(v)}$ carries view-specific semantic information inconsistent with the consensus. Replacing true conditionals with variational classifiers and using the fused posterior $q(\mathbf{z} \mid \{\mathbf{x}\})$:

$$I(\mathbf{y}; \mathbf{z}^{(v)} \mid \mathbf{z}^{\sim(v)}) \leq \mathbb{E}_{p(\{\mathbf{x}\})}\mathbb{E}_{q(\mathbf{z}\mid\{\mathbf{x}\})} \left[ D_{\mathrm{KL}} \left( p_\psi(\mathbf{y} \mid \mathbf{z}) \,\|\, p_\psi(\mathbf{y} \mid \mathbf{z}^{(v)}) \right) \right]. \tag{51}$$

$\square$

This establishes the upper bound in Eq. (9). The corresponding **Classifier Consistency Loss** is:

$$\mathcal{L}_{\text{cls-consist}}^{(v)} = \mathbb{E}_{q(\mathbf{z}\mid\{\mathbf{x}\})} \left[ D_{\mathrm{KL}} \left( p_\psi(\mathbf{y} \mid \mathbf{z}) \,\|\, p_\psi(\mathbf{y} \mid \mathbf{z}^{(v)}) \right) \right], \tag{52}$$

where $\mathbf{z}$ is sampled from the fused posterior $q(\mathbf{z} \mid \{\mathbf{x}\})$ and $\mathbf{z}^{(v)}$ is sampled from $q_\phi(\mathbf{z}^{(v)} \mid \mathbf{x}^{(v)})$.

For multi-label classification where predictions are independent across categories, the KL divergence factorizes:

$$D_{\mathrm{KL}} \left( p_\psi(\mathbf{y} \mid \mathbf{z}) \,\|\, p_\psi(\mathbf{y} \mid \mathbf{z}^{(v)}) \right) = \sum_{c=1}^{C} D_{\mathrm{KL}} \left( p_\psi(y_c \mid \mathbf{z}) \,\|\, p_\psi(y_c \mid \mathbf{z}^{(v)}) \right). \tag{53}$$

For Bernoulli distributions with parameters $\hat{y}_c = \sigma(f_c(\mathbf{z}))$ and $\hat{y}_c^{(v)} = \sigma(f_c(\mathbf{z}^{(v)}))$:

$$D_{\mathrm{KL}} \left( p_\psi(y_c \mid \mathbf{z}) \,\|\, p_\psi(y_c \mid \mathbf{z}^{(v)}) \right) = \hat{y}_c \log \frac{\hat{y}_c}{\hat{y}_c^{(v)}} + (1 - \hat{y}_c) \log \frac{1 - \hat{y}_c}{1 - \hat{y}_c^{(v)}}. \tag{54}$$

### A.6. Summary

Combining the four variational bounds, we obtain the feature-level loss (Eq. (10)):

$$\mathcal{L}_{\text{feat}}^{(v)} = \underbrace{-\mathbb{E}_{q_\phi(\mathbf{z}^{(v)}\mid\mathbf{x}^{(v)})} \left[ \log p_\theta(\mathbf{x}^{(v)} \mid \mathbf{z}^{(v)}) \right]}_{\text{Reconstruction}} + \alpha \underbrace{D_{\mathrm{KL}} \left( q(\mathbf{z} \mid \{\mathbf{x}\}) \,\|\, q_\phi(\mathbf{z} \mid \mathbf{x}^{(v)}) \right)}_{\text{Encoder Consistency}}, \tag{55}$$

and the label-level loss (Eq. (11)):

$$\mathcal{L}_{\text{label}}^{(v)} = \underbrace{-\mathbb{E}_{q_\phi(\mathbf{z}^{(v)}\mid\mathbf{x}^{(v)})} \left[ \log p_\psi(\mathbf{y} \mid \mathbf{z}^{(v)}) \right]}_{\text{Classification}} + \gamma \underbrace{\mathbb{E}_{q(\mathbf{z}\mid\{\mathbf{x}\})} \left[ D_{\mathrm{KL}} \left( p_\psi(\mathbf{y} \mid \mathbf{z}) \,\|\, p_\psi(\mathbf{y} \mid \mathbf{z}^{(v)}) \right) \right]}_{\text{Classifier Consistency}}. \tag{56}$$

The task-relevant representation learning objective aggregates these losses across all available views (Eq. (12)):

$$\mathcal{L}_{\text{TR}} = \sum_{v \in \mathcal{V}} \left[ \mathcal{L}_{\text{feat}}^{(v)} + \lambda \mathcal{L}_{\text{label}}^{(v)} \right], \tag{57}$$

where $\mathcal{V}$ denotes the set of available views for the current sample. The complete training objective (Eq. (18)) integrates all components:

$$\mathcal{L} = \mathcal{L}_{\text{TR}} + \beta \mathcal{L}_{\text{SA}} + \mathcal{L}_{\text{CE}}, \tag{58}$$

where $\mathcal{L}_{\text{SA}}$ denotes the semantic alignment loss from Section 2.2, $\mathcal{L}_{\text{CE}}$ denotes the aggregated classification loss from Section 2.4, and $\alpha, \gamma, \lambda, \beta$ are hyperparameters balancing the different objectives.

Notably, the overall objective involves two classification-related losses, which serve separate and distinct functions. The classification term within $\mathcal{L}_{\text{label}}^{(v)}$, derived from the label sufficiency principle (Section A.4), supervises predictions from each individual view's latent representation $\mathbf{z}^{(v)}$, ensuring that each view-specific encoder independently retains sufficient discriminative information. In contrast, $\mathcal{L}_{\text{CE}}$ supervises the final aggregated predictions $\hat{\mathbf{y}}_{\text{final}}$ obtained through the confidence-weighted fusion mechanism (Eq. (16)), which incorporates outputs from both category-specific conditional posteriors and individual views. This dual-level supervision strategy ensures that individual views maintain discriminative capacity even when other views are missing, while the fusion mechanism learns to optimally combine heterogeneous predictions for improved robustness.

**Implementation.** All expectations are approximated via Monte Carlo estimation. The outer expectation $\mathbb{E}_{p(\mathbf{x})}[\cdot]$ is approximated by the empirical distribution over the training set and computed as the average over mini-batch samples. The inner expectation $\mathbb{E}_{q_\phi(\mathbf{z}|\mathbf{x})}[\cdot]$ is approximated by sampling from the parameterized encoder distribution using the reparameterization trick (Kingma & Welling, 2014): for a Gaussian encoder $q_\phi(\mathbf{z}^{(v)} \mid \mathbf{x}^{(v)}) = \mathcal{N}(\boldsymbol{\mu}_v, \mathrm{diag}(\boldsymbol{\sigma}_v^2))$, samples are drawn as $\mathbf{z}^{(v)} = \boldsymbol{\mu}_v + \boldsymbol{\sigma}_v \odot \boldsymbol{\epsilon}$ where $\boldsymbol{\epsilon} \sim \mathcal{N}(\mathbf{0}, \mathbf{I})$. The label expectation $\mathbb{E}_{p(\mathbf{y}|\mathbf{x})}[\cdot]$ is computed analytically based on the classifier's predicted probability distribution $p_\psi(\mathbf{y} \mid \mathbf{z})$, or collapses to evaluation at the observed labels in supervised settings. The fused posterior $q(\mathbf{z} \mid \{\mathbf{x}\})$ used in the consistency terms is computed via Product-of-Experts fusion as detailed in Section 2.5, which naturally accommodates missing views by restricting the product to available view posteriors.

## B. Experimental Details

### B.1. Datasets

Our experiments span six established benchmarks and two application-oriented datasets. Table 3 summarizes key statistics.

**Benchmark Datasets.** We adopt five image collections with diverse annotation granularity: **Corel5k** (4,999 images, 260 labels), **ESPGame** (20,770 images, 268 labels from crowdsourced tagging), **IAPRTC12** (19,627 images, 291 semantic categories), **Mirflickr** (25,000 social media images, 38 tags), and **Pascal07** (9,963 images, 20 object classes). Each image is represented by six visual descriptors capturing complementary aspects: color distribution (DenseHue, HSV, RGB, LAB histograms), local patterns (DenseSift), and global structure (GIST). We also include **OBJECT**, a multi-attribute recognition dataset with 6,047 instances described by five shape and texture features: color histogram (CH), color moments (CM), autocorrelation (CORR), edge direction histogram (EDH), and wavelet texture (WT).

**NBA Dataset.** Player performance records are collected from Basketball-Reference, yielding 16,992 player-season samples across 2002–2022. We organize statistics into six thematic views: *Scoring* captures shooting volume and efficiency; *Rebounding* reflects physical presence and durability; *Playmaking* encompasses passing, defense, and ball security; *Advanced* aggregates composite efficiency ratings; *Background* encodes player age and team affiliation; *Context* indicates season year and playoff participation. The label space spans three prediction tasks: 10 award indicators (All-Star, All-NBA, MVP candidacy, etc.), 5 positional roles, and 3 career phases (early, prime, late).

**CheXpert-MV Dataset.** To construct a clinically-motivated multi-view benchmark, we process the CheXpert repository (Irvin et al., 2019) as follows: original DICOM files are compressed to JPEG format, and we retain only the 31,413 studies containing paired frontal-lateral acquisitions. Each projection constitutes one view, with visual representations obtained via DenseNet-121 (Huang et al., 2017) pretrained on ImageNet (final classification layer removed), producing 1,024-dimensional embeddings per view. Annotations cover 14 radiological findings: *Atelectasis*, *Cardiomegaly*, *Consolidation*, *Edema*, *Enlarged Cardiomediastinum*, *Fracture*, *Lung Lesion*, *Lung Opacity*, *Pleural Effusion*, *Pleural Other*, *Pneumonia*, *Pneumothorax*, *Support Devices*, and *No Finding*.

*Table 3.* Brief Information of Dataset Statistics.

| Dataset | Domain | #Inst. | #Views | #Labels | Features (Dim.) |
|---|---|---|---|---|---|
| Corel5k | Image | 4,999 | 6 | 260 | |
| ESPGame | Image | 20,770 | 6 | 268 | DenseHue (100), DenseSift (1000), GIST (512), |
| IAPRTC12 | Image | 19,627 | 6 | 291 | HSV (4096), RGB (4096), LAB (4096) |
| Mirflickr | Image | 25,000 | 6 | 38 | |
| Pascal07 | Image | 9,963 | 6 | 20 | |
| OBJECT | Image | 6,047 | 5 | 31 | CH (64), CM (225), CORR (144), EDH (73), WT(128) |
| NBA | Sports | 16,992 | 6 | 18 | Scoring (20), Rebounding (14), Playmaking (15), Advanced (10), Background (41), Context (22) |
| CheXpert-MV | Medical | 31,413 | 2 | 14 | Frontal (1024), Lateral (1024) |

*Table 4.* Experimental results of ten methods on the six datasets with 90% PER and 90% LMR. "AVE" refers to the mean ranking of the corresponding method across all six metrics.

| DATA | METRIC | AIMNet | DICNet | DIMC | iMVWL | LMVCAT | LVSL | MTD | SIP | TDLSR | LGRL |
|---|---|---|---|---|---|---|---|---|---|---|---|
| **COR** | 1-HL | $0.987_{0.001}$ | $0.987_{0.000}$ | $0.987_{0.001}$ | $0.976_{0.001}$ | $0.987_{0.001}$ | $0.987_{0.000}$ | $0.987_{0.001}$ | $0.986_{0.001}$ | $0.987_{0.000}$ | $0.987_{0.000}$ |
| | 1-OE | $0.277_{0.012}$ | $0.242_{0.009}$ | $0.239_{0.011}$ | $0.181_{0.005}$ | $0.229_{0.008}$ | $0.247_{0.006}$ | $0.274_{0.012}$ | $0.289_{0.015}$ | $0.374_{0.010}$ | $0.410_{0.014}$ |
| | 1-Cov | $0.605_{0.005}$ | $0.515_{0.007}$ | $0.518_{0.006}$ | $0.524_{0.004}$ | $0.600_{0.005}$ | $0.608_{0.004}$ | $0.573_{0.006}$ | $0.601_{0.007}$ | $0.692_{0.004}$ | $0.731_{0.005}$ |
| | 1-RL | $0.823_{0.003}$ | $0.774_{0.004}$ | $0.772_{0.005}$ | $0.762_{0.004}$ | $0.817_{0.004}$ | $0.823_{0.001}$ | $0.809_{0.005}$ | $0.821_{0.004}$ | $0.866_{0.002}$ | $0.887_{0.003}$ |
| | AP | $0.240_{0.004}$ | $0.208_{0.005}$ | $0.206_{0.004}$ | $0.163_{0.004}$ | $0.214_{0.001}$ | $0.228_{0.004}$ | $0.234_{0.005}$ | $0.242_{0.006}$ | $0.323_{0.004}$ | $0.360_{0.005}$ |
| | AUC | $0.826_{0.003}$ | $0.776_{0.004}$ | $0.774_{0.004}$ | $0.766_{0.000}$ | $0.820_{0.003}$ | $0.827_{0.003}$ | $0.811_{0.004}$ | $0.823_{0.003}$ | $0.869_{0.002}$ | $0.889_{0.003}$ |
| | AVE | 3.333 | 6.833 | 7.333 | 8.333 | 5.667 | 3.667 | 5.167 | 3.667 | 1.833 | 1.000 |
| **ESP** | 1-HL | $0.982_{0.000}$ | $0.982_{0.000}$ | $0.982_{0.000}$ | $0.969_{0.001}$ | $0.982_{0.000}$ | $0.983_{0.000}$ | $0.982_{0.000}$ | $0.982_{0.000}$ | $0.982_{0.000}$ | $0.983_{0.000}$ |
| | 1-OE | $0.310_{0.007}$ | $0.289_{0.012}$ | $0.283_{0.006}$ | $0.204_{0.001}$ | $0.266_{0.023}$ | $0.265_{0.004}$ | $0.302_{0.008}$ | $0.327_{0.008}$ | $0.385_{0.010}$ | $0.435_{0.012}$ |
| | 1-Cov | $0.508_{0.003}$ | $0.464_{0.000}$ | $0.456_{0.004}$ | $0.421_{0.007}$ | $0.468_{0.003}$ | $0.489_{0.001}$ | $0.492_{0.005}$ | $0.500_{0.004}$ | $0.576_{0.002}$ | $0.589_{0.005}$ |
| | 1-RL | $0.792_{0.001}$ | $0.773_{0.001}$ | $0.769_{0.002}$ | $0.729_{0.004}$ | $0.771_{0.000}$ | $0.783_{0.001}$ | $0.786_{0.000}$ | $0.785_{0.001}$ | $0.825_{0.001}$ | $0.835_{0.001}$ |
| | AP | $0.222_{0.003}$ | $0.210_{0.002}$ | $0.207_{0.003}$ | $0.155_{0.004}$ | $0.201_{0.006}$ | $0.204_{0.001}$ | $0.219_{0.004}$ | $0.225_{0.002}$ | $0.271_{0.003}$ | $0.297_{0.005}$ |
| | AUC | $0.797_{0.001}$ | $0.777_{0.000}$ | $0.772_{0.002}$ | $0.733_{0.005}$ | $0.775_{0.002}$ | $0.787_{0.000}$ | $0.790_{0.000}$ | $0.790_{0.002}$ | $0.830_{0.001}$ | $0.840_{0.001}$ |
| | AVE | 3.167 | 5.833 | 7.000 | 8.667 | 6.833 | 5.833 | 4.167 | 3.500 | 2.000 | 1.000 |
| **IAP** | 1-HL | $0.980_{0.000}$ | $0.980_{0.000}$ | $0.980_{0.000}$ | $0.966_{0.001}$ | $0.980_{0.000}$ | $0.980_{0.000}$ | $0.980_{0.001}$ | $0.980_{0.000}$ | $0.980_{0.000}$ | $0.980_{0.000}$ |
| | 1-OE | $0.342_{0.004}$ | $0.330_{0.009}$ | $0.318_{0.002}$ | $0.245_{0.011}$ | $0.290_{0.007}$ | $0.294_{0.004}$ | $0.344_{0.007}$ | $0.355_{0.005}$ | $0.397_{0.009}$ | $0.463_{0.011}$ |
| | 1-Cov | $0.521_{0.002}$ | $0.472_{0.004}$ | $0.468_{0.002}$ | $0.438_{0.009}$ | $0.471_{0.005}$ | $0.496_{0.002}$ | $0.510_{0.006}$ | $0.519_{0.003}$ | $0.616_{0.001}$ | $0.641_{0.005}$ |
| | 1-RL | $0.818_{0.003}$ | $0.799_{0.004}$ | $0.795_{0.002}$ | $0.761_{0.005}$ | $0.793_{0.003}$ | $0.808_{0.002}$ | $0.807_{0.001}$ | $0.817_{0.003}$ | $0.860_{0.001}$ | $0.876_{0.002}$ |
| | AP | $0.229_{0.002}$ | $0.222_{0.004}$ | $0.215_{0.003}$ | $0.167_{0.003}$ | $0.202_{0.003}$ | $0.208_{0.002}$ | $0.232_{0.001}$ | $0.235_{0.004}$ | $0.278_{0.003}$ | $0.324_{0.005}$ |
| | AUC | $0.822_{0.002}$ | $0.801_{0.003}$ | $0.798_{0.002}$ | $0.766_{0.005}$ | $0.797_{0.001}$ | $0.811_{0.001}$ | $0.811_{0.001}$ | $0.820_{0.002}$ | $0.861_{0.001}$ | $0.878_{0.002}$ |
| | AVE | 3.333 | 5.500 | 6.500 | 8.500 | 7.333 | 5.500 | 4.167 | 3.167 | 1.833 | 1.000 |
| **MIR** | 1-HL | $0.875_{0.001}$ | $0.879_{0.001}$ | $0.877_{0.002}$ | $0.827_{0.004}$ | $0.865_{0.004}$ | $0.874_{0.001}$ | $0.880_{0.001}$ | $0.875_{0.002}$ | $0.885_{0.002}$ | $0.885_{0.002}$ |
| | 1-OE | $0.506_{0.023}$ | $0.533_{0.005}$ | $0.511_{0.005}$ | $0.406_{0.023}$ | $0.470_{0.020}$ | $0.485_{0.004}$ | $0.535_{0.008}$ | $0.540_{0.009}$ | $0.612_{0.006}$ | $0.678_{0.008}$ |
| | 1-Cov | $0.598_{0.006}$ | $0.594_{0.003}$ | $0.589_{0.001}$ | $0.530_{0.012}$ | $0.581_{0.002}$ | $0.584_{0.001}$ | $0.606_{0.006}$ | $0.604_{0.006}$ | $0.645_{0.005}$ | $0.658_{0.004}$ |
| | 1-RL | $0.827_{0.005}$ | $0.828_{0.001}$ | $0.823_{0.001}$ | $0.765_{0.011}$ | $0.817_{0.004}$ | $0.819_{0.002}$ | $0.834_{0.003}$ | $0.830_{0.002}$ | $0.861_{0.003}$ | $0.874_{0.002}$ |
| | AP | $0.494_{0.017}$ | $0.512_{0.001}$ | $0.501_{0.002}$ | $0.415_{0.009}$ | $0.485_{0.010}$ | $0.482_{0.001}$ | $0.519_{0.005}$ | $0.519_{0.001}$ | $0.575_{0.002}$ | $0.609_{0.007}$ |
| | AUC | $0.820_{0.003}$ | $0.823_{0.001}$ | $0.818_{0.001}$ | $0.769_{0.007}$ | $0.808_{0.004}$ | $0.816_{0.000}$ | $0.827_{0.002}$ | $0.823_{0.001}$ | $0.852_{0.000}$ | $0.861_{0.002}$ |
| | AVE | 5.667 | 4.500 | 5.833 | 9.333 | 8.167 | 7.500 | 3.000 | 3.833 | 1.833 | 1.000 |
| **OBJ** | 1-HL | $0.937_{0.001}$ | $0.938_{0.001}$ | $0.938_{0.000}$ | $0.882_{0.005}$ | $0.927_{0.002}$ | $0.934_{0.001}$ | $0.938_{0.000}$ | $0.937_{0.001}$ | $0.946_{0.001}$ | $0.944_{0.001}$ |
| | 1-OE | $0.468_{0.005}$ | $0.453_{0.009}$ | $0.439_{0.005}$ | $0.335_{0.036}$ | $0.405_{0.019}$ | $0.364_{0.004}$ | $0.474_{0.007}$ | $0.485_{0.006}$ | $0.603_{0.020}$ | $0.654_{0.016}$ |
| | 1-Cov | $0.727_{0.009}$ | $0.720_{0.008}$ | $0.709_{0.010}$ | $0.657_{0.023}$ | $0.705_{0.017}$ | $0.712_{0.002}$ | $0.740_{0.007}$ | $0.727_{0.006}$ | $0.795_{0.003}$ | $0.790_{0.008}$ |
| | 1-RL | $0.829_{0.003}$ | $0.823_{0.006}$ | $0.814_{0.008}$ | $0.768_{0.012}$ | $0.806_{0.011}$ | $0.811_{0.001}$ | $0.835_{0.003}$ | $0.828_{0.004}$ | $0.881_{0.002}$ | $0.880_{0.005}$ |
| | AP | $0.506_{0.010}$ | $0.502_{0.006}$ | $0.489_{0.010}$ | $0.394_{0.026}$ | $0.476_{0.009}$ | $0.446_{0.001}$ | $0.519_{0.010}$ | $0.522_{0.011}$ | $0.624_{0.011}$ | $0.647_{0.012}$ |
| | AUC | $0.842_{0.003}$ | $0.836_{0.005}$ | $0.828_{0.008}$ | $0.784_{0.012}$ | $0.821_{0.011}$ | $0.827_{0.001}$ | $0.848_{0.003}$ | $0.840_{0.004}$ | $0.891_{0.001}$ | $0.888_{0.005}$ |
| | AVE | 4.333 | 5.333 | 6.333 | 9.333 | 8.000 | 7.500 | 3.333 | 4.000 | 1.333 | 1.667 |
| **PAS** | 1-HL | $0.923_{0.003}$ | $0.927_{0.001}$ | $0.927_{0.001}$ | $0.871_{0.003}$ | $0.921_{0.003}$ | $0.926_{0.000}$ | $0.926_{0.001}$ | $0.923_{0.000}$ | $0.927_{0.003}$ | $0.932_{0.001}$ |
| | 1-OE | $0.382_{0.023}$ | $0.402_{0.002}$ | $0.403_{0.002}$ | $0.306_{0.037}$ | $0.376_{0.021}$ | $0.415_{0.001}$ | $0.395_{0.011}$ | $0.389_{0.011}$ | $0.415_{0.016}$ | $0.439_{0.015}$ |
| | 1-Cov | $0.658_{0.002}$ | $0.636_{0.013}$ | $0.626_{0.013}$ | $0.589_{0.014}$ | $0.630_{0.018}$ | $0.654_{0.005}$ | $0.674_{0.005}$ | $0.668_{0.018}$ | $0.753_{0.006}$ | $0.768_{0.007}$ |
| | 1-RL | $0.727_{0.006}$ | $0.710_{0.011}$ | $0.703_{0.010}$ | $0.658_{0.014}$ | $0.693_{0.016}$ | $0.726_{0.004}$ | $0.740_{0.003}$ | $0.729_{0.013}$ | $0.808_{0.007}$ | $0.824_{0.006}$ |
| | AP | $0.440_{0.009}$ | $0.440_{0.005}$ | $0.434_{0.004}$ | $0.368_{0.021}$ | $0.430_{0.005}$ | $0.444_{0.002}$ | $0.447_{0.009}$ | $0.447_{0.008}$ | $0.449_{0.009}$ | $0.524_{0.010}$ |
| | AUC | $0.754_{0.003}$ | $0.737_{0.011}$ | $0.727_{0.012}$ | $0.690_{0.013}$ | $0.727_{0.010}$ | $0.754_{0.003}$ | $0.766_{0.006}$ | $0.761_{0.013}$ | $0.831_{0.006}$ | $0.845_{0.005}$ |
| | AVE | 5.333 | 5.333 | 6.000 | 8.667 | 7.500 | 4.500 | 3.500 | 4.167 | 2.000 | 1.000 |

## B.2. Comparison Methods

We benchmark against nine representative methods spanning matrix factorization, deep learning, and information-theoretic approaches. **iMVWL** (Tan et al., 2018) recovers missing views via low-rank matrix completion. **LVSL** (Zhao et al., 2022) disentangles view-specific and label-specific representations without handling missingness. Among deep methods, **DICNet** (Liu et al., 2023b) enforces instance-level consistency via contrastive objectives; **DIMC** (Wen et al., 2023) imposes cross-view alignment through consistency regularization; **LMVCAT** (Liu et al., 2023c) aggregates views using cross-attention transformers. Recent advances include **AIMNet** (Liu et al., 2024a) with attention-guided imputation, **MTD** (Liu et al., 2023a) employing masked autoencoders for robust reconstruction, and **SIP** (Liu et al., 2024b) optimizing an information bottleneck objective. **TDLSR** (Li et al., 2025) constructs view-specific topology and prototype association graphs to derive label-specific representations, and provides theoretical analysis of generalization performance.

## B.3. Evaluation Metrics

Performance is assessed via six standard multi-label metrics. **Hamming Loss (HL)** quantifies the proportion of misclassified instance-label pairs. **Ranking Loss (RL)** measures the fraction of reversely ordered positive-negative label pairs. **OneError**

(OE) indicates whether the highest-ranked prediction is incorrect. **Coverage (Cov)** counts how far down the ranked list one must traverse to cover all ground-truth labels, normalized by the number of labels. **Average Precision (AP)** averages the precision at each positive label's rank. **Area Under Curve (AUC)** estimates the probability of ranking a random positive above a random negative. Lower values are better for HL, RL, OE, and Cov; higher values are better for AP and AUC. For uniform interpretation, we report 1-HL, 1-RL, 1-OE, and 1-Cov so that higher is consistently better.

### B.4. Implementation Details

Data are partitioned into training, validation, and test splits at a 7:1:2 ratio. Incomplete scenarios are simulated by randomly masking view instances according to the Partial Example Ratio (PER), with the constraint that every sample retains at least one observed view. Label annotations are independently dropped following the Label Missing Ratio (LMR). Masking patterns are fixed across methods to ensure fair comparison.

Optimization uses Adam with learning rate $10^{-4}$. The contrastive temperature $\tau$ is set to 0.1. All other hyperparameters, including the encoder consistency weight $\alpha$, classifier consistency weight $\gamma$, confidence aggregation temperature $\kappa$, feature-label trade-off $\lambda$, and semantic alignment weight $\beta$, are fixed at 1. Experiments run on an NVIDIA RTX 4090 (24GB), with results averaged over 10 independent runs for the $50\%$ PER and $50\%$ LMR setting, and over 5 independent runs for all other missing rate combinations.

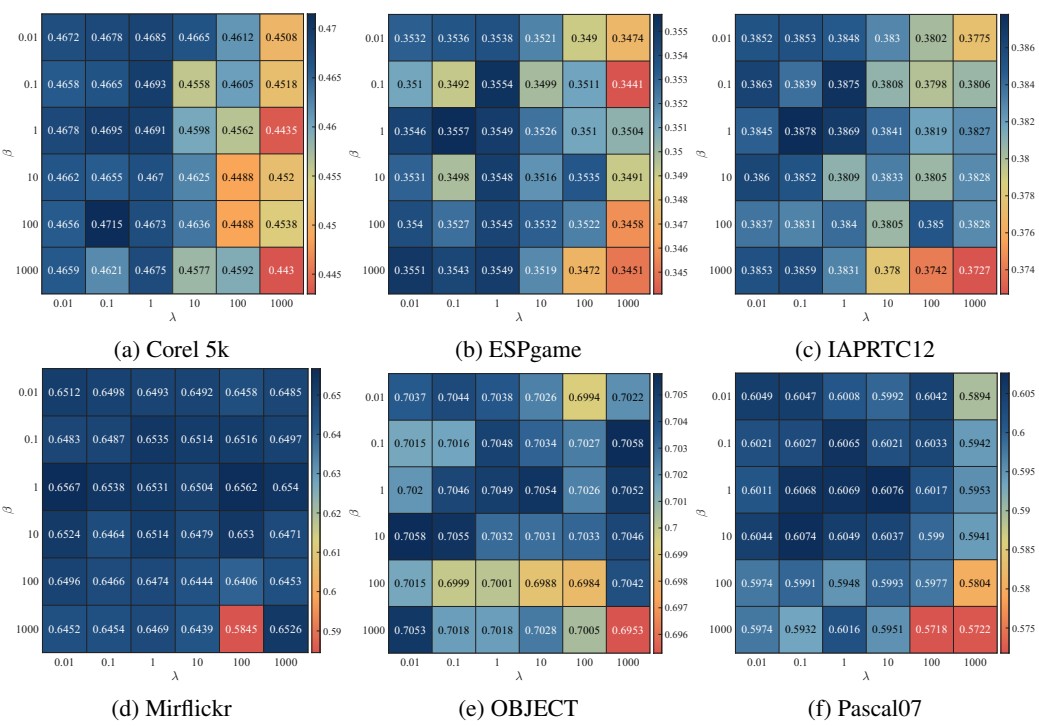

*Figure 5.* Parameter analysis of the trade-off parameters $\lambda$ and $\beta$.

### B.5. Results

**Parameter Sensitivity.** The hyperparameters $\lambda$ and $\beta$ control the trade-off between feature-level and label-level objectives, and the influence of semantic alignment, respectively. We conduct sensitivity analysis by selecting both parameters from the range $\{0.01, 0.1, 1, 10, 100\}$ and present the joint influence on AP in the heatmaps shown in Fig. 5. The results indicate that LGRL achieves relatively stable performance when both $\lambda$ and $\beta$ fall within the range of $(0.1, 1)$, demonstrating low sensitivity to hyperparameter selection. Moreover, the optimal configurations vary slightly across datasets, suggesting that the balance between different loss components may depend on dataset characteristics such as label space size and view heterogeneity.

**Complexity Analysis.** To characterize the computational cost, we note that prototype learning introduces $\mathcal{O}(C \cdot d_z)$ parameters, Product-of-Experts fusion requires $\mathcal{O}(|\mathcal{V}| \cdot d_z)$ operations per category, and confidence-weighted aggregation

adds $\mathcal{O}((1 + |\mathcal{V}|) \cdot C)$ operations. Table 5 compares the runtime across methods. LGRL incurs moderate overhead due to category-specific posterior computation, but remains competitive with existing methods.

*Table 5.* Runtime comparison (seconds) on four datasets.

| Method | Corel5k | Pascal07 | Espgame | Mirflickr |
|---|---|---|---|---|
| iMVWL | 513.41 | 356.28 | 1822.23 | 568.46 |
| AIMNet | 408.37 | 247.15 | 607.02 | 159.98 |
| DICNet | 765.02 | 489.63 | 3047.17 | 1228.27 |
| DIMC | 559.77 | 372.41 | 995.89 | 359.66 |
| LMVCAT | 1198.72 | 723.56 | 3037.25 | 151.27 |
| MTD | 2520.99 | 1456.82 | 4080.72 | 562.55 |
| SIP | 1039.93 | 583.47 | 1605.16 | 727.26 |
| TDLSR | 2270.64 | 1327.14 | 3882.59 | 520.24 |
| LGRL | 1287.35 | 591.23 | 1738.42 | 812.53 |

**Performance under Extreme Incompleteness.** To evaluate the robustness of LGRL under severe data scarcity, we conduct experiments with 90% partial example ratio (PER) and 90% label missing ratio (LMR), representing an extreme scenario where only 10% of views and 10% of labels are observed. Table 4 presents the comparison results across six benchmark datasets. Despite the challenging setting, LGRL achieves the best overall performance on five of the six datasets, attaining an average ranking of 1, and remains highly competitive on OBJECT (ranking second to TDLSR). Notably, the performance gap between LGRL and competing methods becomes more pronounced under this extreme setting. For instance, on Corel5k, LGRL achieves an AP of 0.360 compared to 0.323 for the second-best method TDLSR, representing an 11.5% relative improvement, and a 48.8% gain over SIP (0.242). Similar trends are observed on other datasets, validating that the label-guided representation learning strategy provides substantial benefits when supervision signals are extremely sparse.

**Impact of Varying Missing Ratios.** To further investigate the performance under different incompleteness levels, we vary the partial example ratio (PER) from 50% to 90% while fixing the label missing ratio (LMR) at 50%, 70%, and 90%, respectively. Figures 6, 7, and 8 present the radar charts of ten methods across six metrics on five datasets under these settings. Additionally, Figures 9, 10, 11, and 12 show the AP, AUC, 1-RL, and 1-OE results on four datasets with both PER and LMR synchronously varying from 30% to 90%.

**Application to Real-World Scenarios.** Figure 13 presents the radar charts illustrating the performance of ten methods on the NBA and CheXpert-MV datasets under varying missing ratios. On the NBA dataset, LGRL consistently achieves competitive performance across all six metrics, and the advantage becomes more pronounced as the missing ratio increases from 30% to 90%. On the CheXpert-MV dataset, which contains only two views derived from medical imaging, LGRL demonstrates its effectiveness in learning discriminative representations under various incompleteness levels. These results confirm that LGRL generalizes well to diverse real-world applications with different data characteristics.

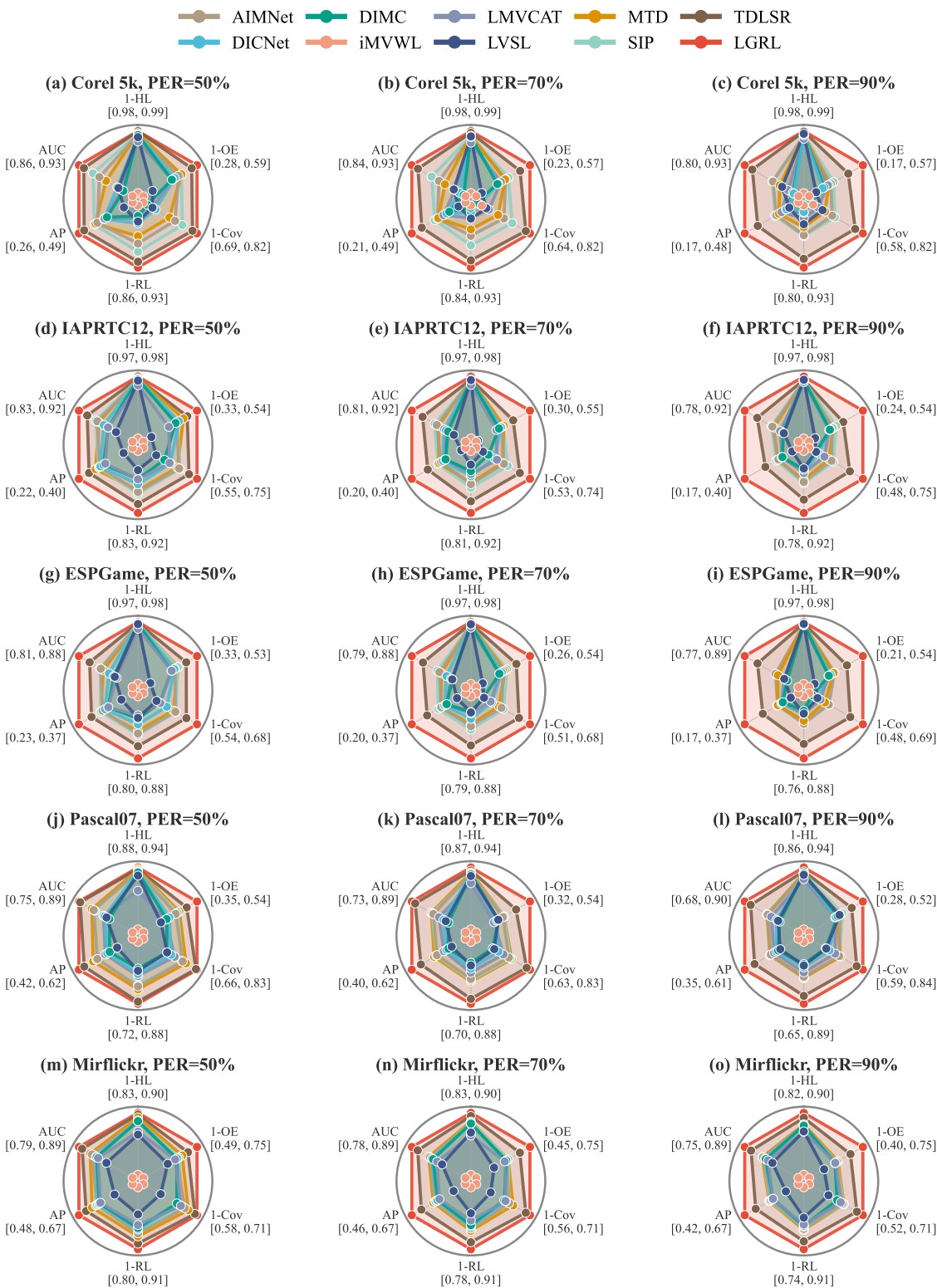

*Figure 6.* Experimental results of ten methods on five datasets with PER varying from 50% to 90% while LMR = 50%.

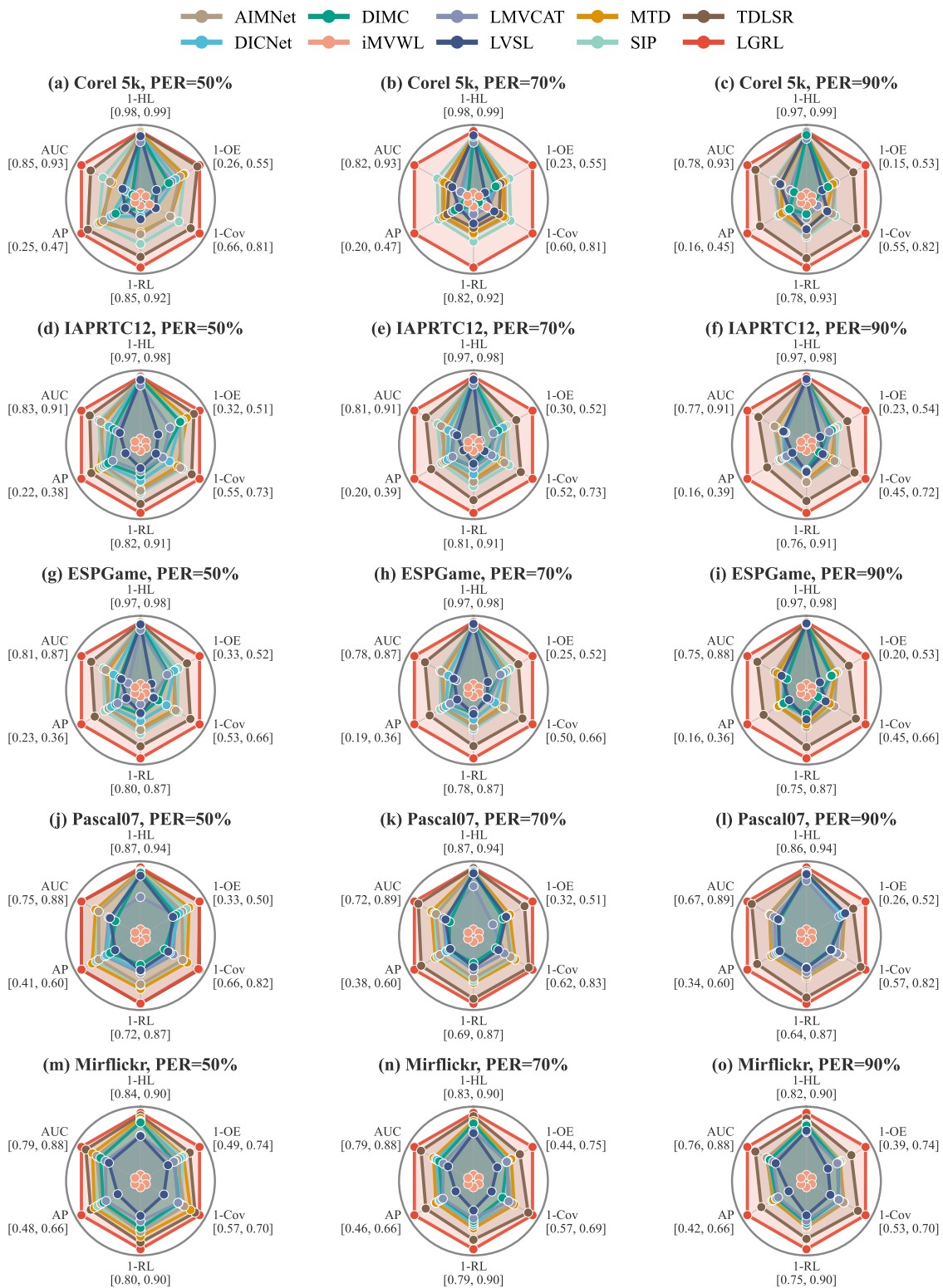

*Figure 7.* Experimental results of ten methods on five datasets with PER varying from 50% to 90% while LMR = 70%.

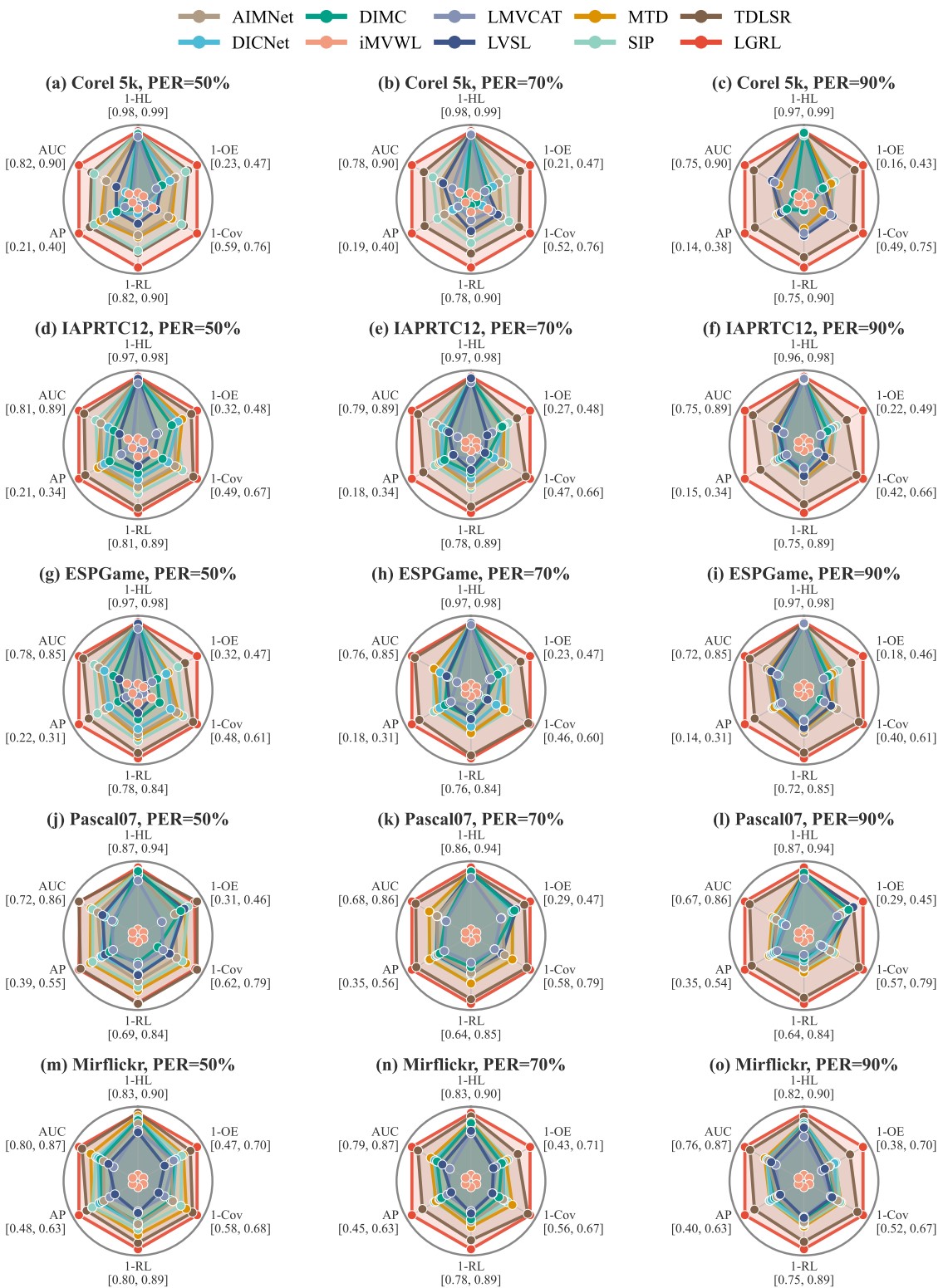

*Figure 8.* Experimental results of ten methods on five datasets with PER varying from 50% to 90% while LMR = 90%.

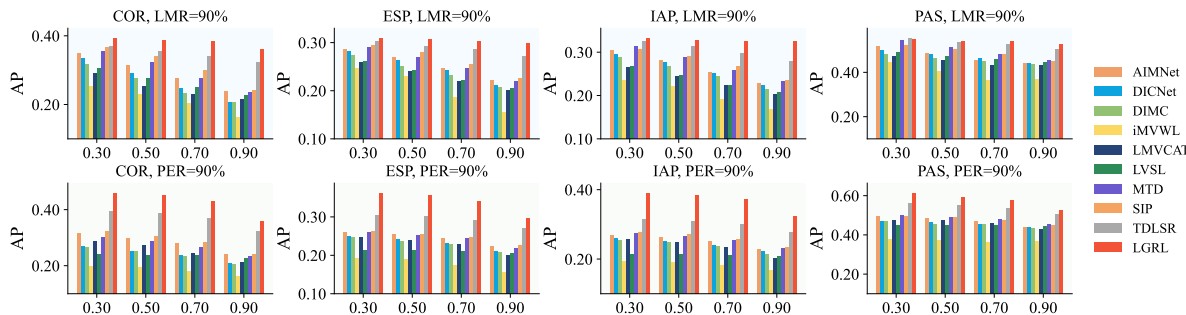

*Figure 9.* AP results on four datasets with PER and LMR changing from 30% to 90%.

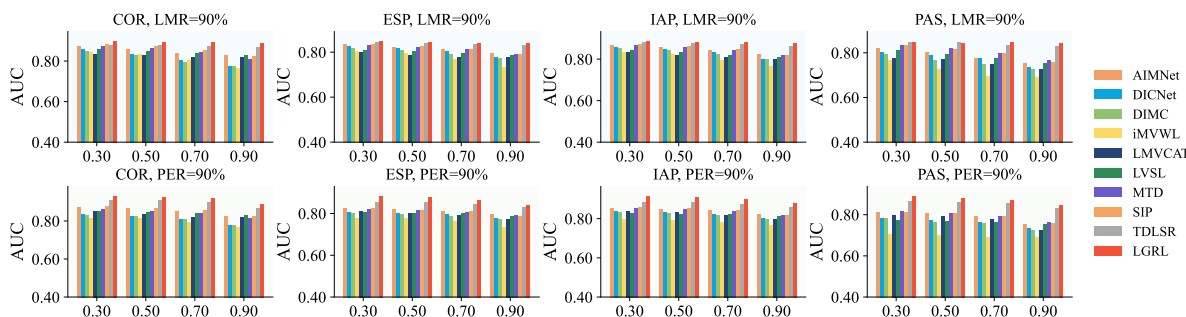

*Figure 10.* AUC results on four datasets with PER and LMR changing from 30% to 90%.

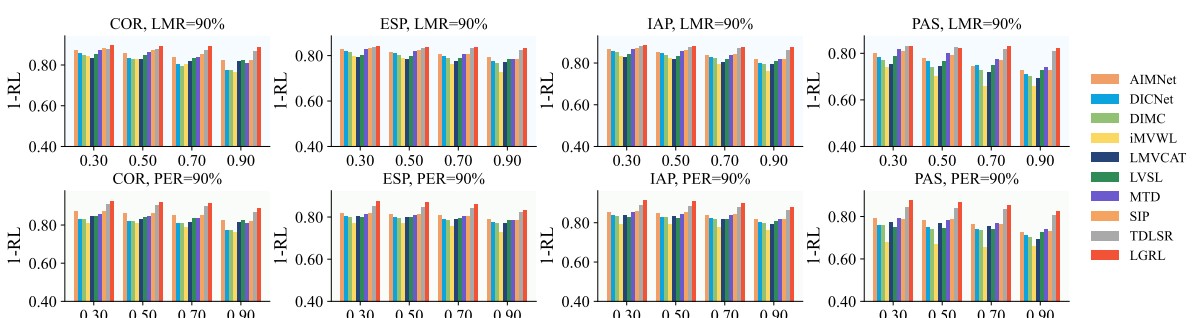

*Figure 11.* 1-RL results on four datasets with PER and LMR changing from 30% to 90%.

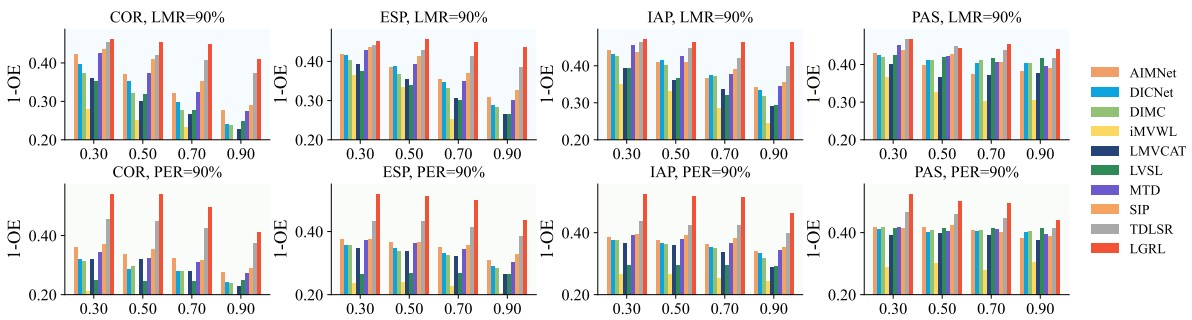

*Figure 12.* 1-OE results on four datasets with PER and LMR changing from 30% to 90%.

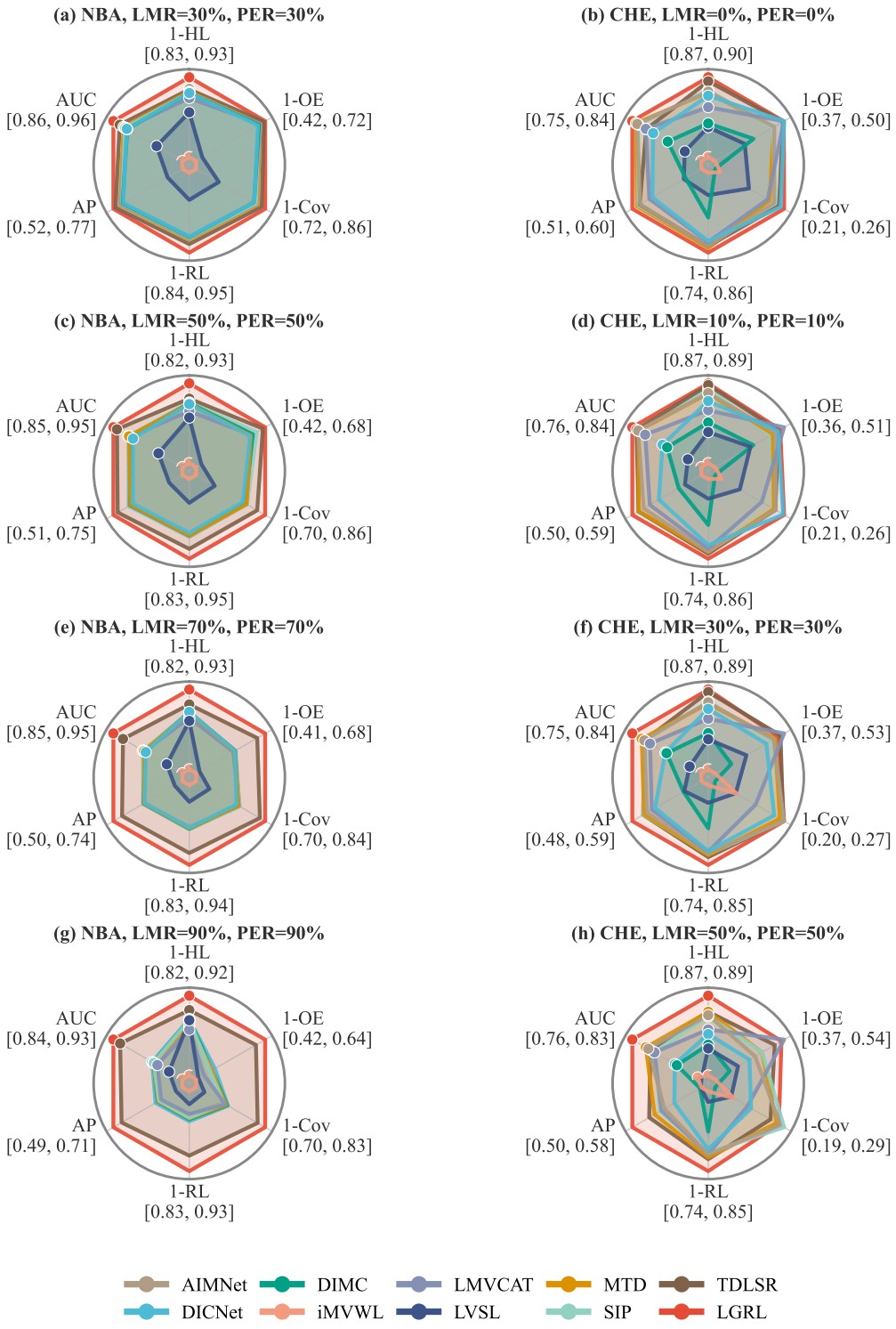

*Figure 13.* Radar charts of ten methods on two application datasets. The left column shows results on the NBA dataset with LMR and PER synchronously varying from 30% to 90%, while the right column presents results on the CheXpert-MV dataset with LMR and PER synchronously varying from 0% to 50%. Each axis represents one evaluation metric, and the range is annotated in brackets.

