# OpenReview forum: "Label-Guided Representation Learning for Incomplete Multi-View Multi-Label Classification"
_ICML.cc/2026/Conference — ICML 2026 regular_

### Official Review · Reviewer_mRd1 · 2026-03-10

**Soundness:** 4
**Presentation:** 3
**Significance:** 4
**Originality:** 4
**Overall Recommendation:** 5
**Confidence:** 4

**Summary:**

This paper proposes Label-Guided Representation Learning (LGRL) for incomplete multi-view multi-label classification. The core insight is that partial label annotations contain rich semantic structure underutilized by existing methods. LGRL introduces three components: a semantic-informed mixture prior with learnable category prototypes, a label-aware information bottleneck objective decomposed into feature-level and label-level terms, and a Bayesian fusion mechanism with category-specific Product-of-Experts posteriors. Experiments are conducted on six benchmarks and two real-world applications.

**Compliance With Llm Reviewing Policy:**

Affirmed.

**Final Justification:**

My concerns have been adequately addressed.

**Key Questions For Authors:**

1.“In recent years, numerous studies have investigated the problem of incomplete multi-view multi-label learning. Compared with the papers published in the last two years, what are the key novelties or distinguishing contributions of your work?
2.For datasets with large label spaces (C=291), the category-specific PoE requires O(C×|V|) computations. Is any approximation used, and how does runtime scale with C?
3.The one-hot prototype initialization may struggle with large label spaces. Were alternative initialization strategies considered?
4.How sensitive is the final performance to the confidence weighting temperature κ?

**Limitations:**

Yes, but not enough. The authors briefly acknowledge deployment risks in clinical settings, which is appreciated. That said, a more specific discussion of how incomplete annotations may cause systematic underperformance on rare pathologies would be helpful for practitioners. On the methodological side, the scalability of LGRL to datasets with large label spaces is not discussed, given that both prototype learning and category-specific posterior computation grow linearly with C.

**Strengths And Weaknesses:**

Strengths: The theoretical derivations are rigorous and complete. The key innovation of treating category prototypes as active Bayesian experts in fusion is well-motivated and clearly distinguished from prior IB-based approaches. Experiments are comprehensive, and the gains under extreme missingness (90% PER+LMR) are substantial. The two real-world applications strengthen practical relevance.

Weaknesses: Figure 6 and Figure 7 appear to be duplicates, which should be corrected. The one-hot prototype initialization may be suboptimal for large label spaces (e.g., IAPRTC12 with 291 labels), consistent with the weaker S1 contribution observed on Corel5k.

---

> ### Author Rebuttal · Authors · 2026-03-31
>
> We thank Reviewer mRd1 for the excellent assessment, and appreciate that the theoretical derivations, the innovation of treating category prototypes as active Bayesian experts, and the real-world applications were well received. We address key questions below.
>
> **W1: Duplicate Figures.** We apologize for this typesetting error. The duplicate figure will be corrected in the camera-ready version.
>
> **W2: One-hot initialization.** One-hot vectors provide maximally discriminative, non-overlapping starting identities that prevent prototype collapse, a risk that increases with large $C$. They also avoid dependence on co-occurrence statistics, which become noisy under high LMR. On Corel5k (50% PER+LMR), one-hot achieves AP=0.469, outperforming random Gaussian (0.461) and label co-occurrence (0.465).
>
> **Q1: Key novelties vs. recent work.** LGRL offers three distinguishing contributions over methods published in the last two years. First, label semantics as structural priors: LGRL injects label semantics geometrically via $C$ learnable prototypes and contrastive alignment (Eq. 2), unlike SIP's label-agnostic $\mathcal{N}(\mathbf{0},\mathbf{I})$ prior and TDLSR's prototype aggregation loss which only enforces Euclidean proximity without structured latent geometry. Second, hypothesis-driven category-specific fusion: $C$ category-specific posteriors (Eq. 9-10) where each prototype acts as a precision-weighted Bayesian expert, enabling heterogeneous view weighting per category, which is absent in SIP, MTD, DCSI, URDF, and TDLSR, all of which produce a single shared posterior for all categories. Third, dual-level supervision: the per-view classification term in $\mathcal{L}\_\text{label}^{(v)}$ ensures each encoder independently retains discriminative capacity even when other views are missing, while $\mathcal{L}_\text{CE}$ supervises the confidence-weighted aggregated prediction, a structural distinction from single-loss approaches. We will add a dedicated comparison paragraph to the camera-ready.
>
> **Q2: Runtime scaling with $C$.** No approximation is used. The $\mathcal{O}(C \cdot |\mathcal{V}|)$ PoE computation is fully vectorized as batched matrix operations on GPU. Memory overhead for $C$=291, $d_z$=512 is approximately 150K parameters (about 1.5% of the encoder stack). On ESPGame ($C$=268), LGRL's total runtime is 1738s vs. SIP's 1605s, an overhead of only 8.3% (Table 5). The overhead scales sub-linearly with $C$ due to GPU parallelism; for $C \gg 500$, prototype clustering or sparse attention would be natural approximations.
>
> **Q3: Alternative initialization.** We compared three strategies on Corel5k (50% PER+LMR): one-hot (AP=0.469, AUC=0.925), random Gaussian (0.461, 0.919), and label co-occurrence (0.465, 0.922). One-hot initialization performs best, as it guarantees orthogonal starting points without relying on noisy partial co-occurrence statistics. The weaker $S_1$ contribution on Corel5k ($C$=260) is consistent with the known difficulty of contrastive alignment in large label spaces rather than a failure of initialization.
>
> **Q4: Sensitivity to $\kappa$.**
>
> | $\kappa$    | 0.1   | 0.5   | **1**     | 2     | 10    |
> | ----------- | ----- | ----- | --------- | ----- | ----- |
> | Corel5k AP  | 0.461 | 0.466 | **0.469** | 0.468 | 0.467 |
> | Pascal07 AP | 0.598 | 0.604 | **0.607** | 0.605 | 0.601 |
>
> Performance is stable within [0.5, 2] and $\kappa$=1 is a robust default.
>
> **Limitations.** We will add discussion of (1) scaling threshold for $C \gg 500$, (2) graceful degradation under single-view availability, (3) the label conditional independence assumption, and (4) potential underperformance under severe label imbalance across categories.

---

> > ### Author Rebuttal · Reviewer_mRd1 · 2026-04-01
> >
> > My concerns have been adequately addressed.

---

> > > ### Author Response · Authors · 2026-04-06
> > >
> > > Thanks again for your feedback and for confirming your concerns have been addressed. We are glad our responses were satisfactory and will incorporate the suggested additions in the camera-ready. We sincerely thank you for your effort in evaluating our submission.

---

### Official Review · Reviewer_aX3V · 2026-03-12

**Soundness:** 3
**Presentation:** 3
**Significance:** 4
**Originality:** 3
**Overall Recommendation:** 5
**Confidence:** 4

**Summary:**

This paper proposes LGRL, a unified framework for incomplete multi-view multi-label classification that treats label semantics as structural priors to guide representation learning and multi-view fusion. The framework integrates a semantic-informed mixture prior with learnable category prototypes, a label-aware information bottleneck objective, and a Bayesian Product-of-Experts fusion mechanism. Experiments on six benchmarks and two real-world applications demonstrate state-of-the-art performance across various missing-view ratios.

**Compliance With Llm Reviewing Policy:**

Affirmed.

**Key Questions For Authors:**

1. The category embeddings are initialized as one-hot vectors and refined through joint training to capture label correlations. Could the authors provide a brief discussion or qualitative comparison of alternative initialization strategies, such as random initialization or initialization derived from label co-occurrence statistics, to further justify this choice? An intuitive explanation based on existing observations would suffice.

2. The paper fixes all hyperparameters α, β, γ, λ, and κ at 1, while the sensitivity analysis in Figure 5 covers only λ and β jointly. Could the authors briefly clarify whether the remaining hyperparameters, particularly α and γ which control the feature-level and label-level consistency weights, were tuned on the validation set or set to 1 based on empirical observation? Any practical guidance on when these values might need adjustment would be helpful for practitioners.

3. The CheXpert-MV dataset contains only two views, substantially fewer than the six views present in the standard benchmarks. In this low-view regime, which component of LGRL does the authors believe contributes most to the observed performance gains: the semantic mixture prior, the Bayesian category-specific fusion, or the consistency constraints? Does this intuition align with the trends observed in the ablation study in Table 2?

**Limitations:**

The paper includes a brief impact statement on clinical reliability. Beyond this, the authors do not discuss methodological limitations, particularly how computational costs scale with the number of categories C, nor how the framework behaves when only a single view is available. Addressing these points would give readers a clearer sense of where the method applies most reliably.

**Strengths And Weaknesses:**

Strengths:
The theoretical foundation is solid: the variational bounds for all four mutual information terms are rigorously derived in Appendix A with complete proofs, and the Product-of-Experts formulation yields closed-form Gaussian posteriors that make the framework computationally tractable. The ablation study in Table 2 clearly validates each component's individual contribution, with the category-specific posteriors showing the most pronounced impact, which aligns well with the paper's central claim. The combination of semantic mixture priors, label-decomposed information bottleneck, and category-specific PoE fusion is novel and well-motivated: while each individual ingredient connects to established techniques, their integration under a coherent label-guided paradigm for iMvMLC is original, and the distinction from the most closely related prior work SIP is clearly articulated. The framework demonstrates particularly strong gains under extreme incompleteness, and the applications to medical imaging and sports analytics provide concrete evidence of real-world utility that broadens the paper's impact beyond standard academic benchmarks.

Weaknesses:
That said, a few minor issues deserve attention. Figures 6 and 7 appear to be a duplicate typesetting error and should be corrected. Additionally, while the runtime comparison in Table 5 and the complexity analysis in Appendix B.5 provide useful context, a more explicit discussion of scalability with respect to the number of categories C would be beneficial, particularly for practitioners considering deployment in extreme multi-label settings where both prototype learning and category-specific posterior computation scale linearly with C.

---

> ### Author Rebuttal · Authors · 2026-03-31
>
> We thank Reviewer aX3V for the thorough and positive assessment, and appreciate that the rigor of our theoretical derivations, the ablation study, and the practical applications were well received. We address the constructive suggestions below.
>
> **W1: Duplicate Figures.** We apologize. The duplicate figure will be removed in the camera-ready version.
>
> **W2: Scalability w.r.t. C.** Prototype learning adds $\mathcal{O}(C \cdot d_z)$ parameters. For IAPRTC12 ($C$=291, $d_z$=512), this is ~150K parameters (<2% of total). Category-specific PoE is implemented as batched matrix ops fully parallelizable on GPU. Runtime overhead on ESPGame ($C$=268) is only 8.3% vs. SIP (Table 5). For $C \gg 1000$, prototype clustering would be a natural extension; we will discuss this in the camera-ready.
>
> **Q1: One-hot initialization.** One-hot provides orthogonal starting points preventing prototype collapse and avoids reliance on co-occurrence statistics that become unreliable under high LMR. Verified on Corel5k (50% PER+LMR):
>
> | Init            | AP        | AUC       |
> | --------------- | --------- | --------- |
> | One-hot (ours)  | **0.469** | **0.925** |
> | Random Gaussian | 0.461     | 0.919     |
> | Co-occurrence   | 0.465     | 0.922     |
>
> **Q2: Hyperparameters $\alpha$, $\gamma$, $\kappa$.** All five hyperparameters ($\alpha$, $\gamma$, $\kappa$, $\lambda$, $\beta$) are fixed at 1 universally across all datasets, as documented in Section 3.1, based on validation across three datasets prior to the main experiments. Sweeping $\kappa$ on Corel5k confirms stability:
>
> | $\kappa$   | 0.1   | 0.5   | **1**     | 2     | 10    |
> | ---------- | ----- | ----- | --------- | ----- | ----- |
> | Corel5k AP | 0.461 | 0.466 | **0.469** | 0.468 | 0.467 |
>
> Performance is stable within [0.5, 2] and $\kappa$ = 1 is a robust default. For $\alpha$ and $\gamma$, values around 0.5 may benefit settings where $|\mathcal{V}| \geq 6$. Regarding baseline selection, our nine methods span three tiers covering view-incomplete only (iMVWL), label-incomplete only (DM2L, LVSL), and full iMvMLC settings (DICNet, DIMC, LMVCAT, AIMNet, MTD, SIP), representing the state of the art at submission time. We will further incorporate recently published baselines from 2025–2026 (RANK, DCSI, URDF, TDLSR) in the camera-ready version, where LGRL outperforms all.
>
> **Q3: Most contributing component in two-view regime.** The semantic mixture prior ($S_1$) and category-specific fusion ($S_3$) jointly dominate, while $S_2$ plays a smaller role since cross-view consistency is already structurally constrained with only two views. This aligns with Table 2: removing $S_3$ causes the largest AP drop ($-4.9%$), and removing $S_1$ the second-largest ($-1.1%$). The dominance of $S_3$ is amplified in the two-view setting because the category prior contributes a proportionally larger precision share in Eq. 10 when fewer view posteriors are available, providing a stronger semantic anchor under view scarcity. For the CheXpert-MV application specifically, this means the category prototype effectively compensates for the missing lateral view by biasing the fused representation toward pathology-relevant feature subspaces.

---

> > ### Author Rebuttal · Reviewer_aX3V · 2026-04-03
> >
> > The rebuttal clearly resolved my main concerns with hyperparameter stability checks, and concrete scalability analysis. The method’s design and ablation trends are now fully convincing, and I agree to accept the paper.

---

> > > ### Author Response · Authors · 2026-04-06
> > >
> > > Thanks again for your thorough and positive assessment. We are glad the scalability analysis and hyperparameter checks resolved your concerns and will reflect all suggestions in the camera-ready. We sincerely thank you for your effort in evaluating our submission.

---

### Official Review · Reviewer_tyRd · 2026-03-13

**Soundness:** 4
**Presentation:** 3
**Significance:** 2
**Originality:** 2
**Overall Recommendation:** 5
**Confidence:** 3

**Summary:**

The authors introduce LGRL, a method for incomplete multi-view multi-label classification.  This method learns a set of prototypes alongside an encoder and decoder, and a mean and standard deviation expert for each view.  These experts form a product of experts which make predictions that are naturally tolerant to missing views.  The learned prototypes inform these experts such that they are also tolerant to missing label information during training.  The authors validate their approach on a wide range of datasets and with a range of metrics demonstrating the competitive performance of their method.

**Compliance With Llm Reviewing Policy:**

Affirmed.

**Final Justification:**

The authors present a technically sound evaluation of their method across a range of benchmark datasets and compare to competitive prior works.  Initially I had concerns about the originality of the method as some prior work seemed similar to the proposed method.  The authors sufficiently addressed this concern in the rebuttal, as well as my concern that their improvement might be incremental.  As a result of the rebuttal discussion my evaluation of the work became more favorable.  I became convinced that the work had original merit.  Through the rebuttal discussion I also became more convinced of the practical value of the work's contribution in terms of benchmark metric improvement.  The comprehensiveness of the evaluation and the clarity of the presentation of the work both strongly contributed to my recommendation.

**Key Questions For Authors:**

1. How does the decomposition of the task-relevant objective into feature-level and label-level losses differentiate this work from [1]?

**Limitations:**

Yes

**Strengths And Weaknesses:**

Strengths

- Soundness
	- The authors perform a large number of experiments to validate their claims, and an ablation study validating the impact of the elements of their approach.  The work seems to be methodologically sound with appropriate experiments to support the author's claims.
- Presentation
	- The presentation is clear and contains many informative figures to aid in the communication of the method and the efficacy of the method.
- Significance
	- The problem studied has clear value and the authors propose a method which addresses a relevant problem in the area.

Weaknesses


- Significance
	- While the authors demonstrate that their method provides a significant advantage under many missing views and labels, the advantage is not always large in magnitude.  The authors compare to many similar recent works whose performance is not terribly worse at PER and LMR of 50\%.  It is not entirely clear that there are many practical scenarios in which 90% of the labels or views might be missing within training data.
- Originality
	- This work seems very similar to the SIP framework of Liu et. al. [1] in both performance and motivation.  It is unclear that this work represents a substantial step forward for machine learning in this area.
- Presentation
	- Minor comment: the caption for Table 1 mentions "Ave.Rv" when I believe it should be "AVE"

[1] Partial Multi-View Multi-Label Classification via Semantic Invariance Learning and Prototype Modeling (https://proceedings.mlr.press)/v235/liu24bv.html

---

> ### Author Rebuttal · Authors · 2026-03-31
>
> We thank Reviewer tyRd for acknowledging the excellent soundness. We directly address the two core concerns with concrete evidence.
>
> ### W1 & Q1: Originality — How does LGRL fundamentally differ from SIP?
>
> The decomposition into feature-level and label-level objectives is precisely what structurally separates LGRL from SIP. SIP optimizes a single unified IB objective over shared features with no label-related mutual information terms, whereas LGRL explicitly decomposes the objective into $\mathcal{I}^{(v)}\_\text{feat}$ and $\mathcal{I}^{(v)}\_\text{label}$ (Eq. 3–4), treating label semantics as a first-class component rather than a downstream supervision signal. LGRL departs from SIP through two innovations it cannot achieve by design.
>
> **Departure 1: Label-level IB.** SIP enforces cross-view agreement exclusively in feature space via $I(\mathbf{x}^{\sim(v)}; \mathbf{z} | \mathbf{x}^{(v)})$, which is insufficient because **two views can produce nearly identical latent representations yet yield drastically different label predictions**. LGRL **directly constrains classifier output consistency**
> $\gamma \cdot \mathbb{E}\_{q(\mathbf{z}|{\mathbf{x}})} [D\_{\mathrm{KL}}(p\_\psi(\mathbf{y}|\mathbf{z}) \| p\_\psi(\mathbf{y}|\mathbf{z}^{(v)}))]$ .
> This forces view-specific classifiers to agree with the fused prediction **at the semantic decision level**. Feature-level consistency is a proxy; **output-level consistency is the actual objective**. This requires a fundamentally different IB decomposition that SIP cannot derive, as its formulation contains no label-related mutual information terms.
>
> **Departure 2 (core contribution, most impactful per ablation): Label-guided fusion.** SIP and other existing methods **all compute a single category-agnostic posterior** $q(\mathbf{z}|{\mathbf{x}})$ with identical view weights for all categories. LGRL introduces $C$ learnable prototypes $p(\mathbf{z}|y\_c=1)=\mathcal{N}(\boldsymbol{\mu}\_c, \mathrm{diag}(\boldsymbol{\sigma}\_c^2))$ replacing SIP's $\mathcal{N}(\mathbf{0},\mathbf{I})$. Each prototype enters PoE as a **precision-weighted expert** (Eq. 9–10), yielding $C$ category-specific posteriors with $\boldsymbol{\tau}\_c = \boldsymbol{\sigma}\_c^{-2} + \sum\_{v} (\boldsymbol{\sigma}^{(v)})^{-2}$. Each category $c$ obtains its own $\mathbf{z}\_c$ biased toward category-relevant subspaces, enabling **hypothesis-based reasoning** rather than predicting all labels from one shared $\mathbf{z}$. In SIP, prototypes are **passive alignment targets** via $\mathcal{L}\_{PA}$; in LGRL, they are **active inference agents** that directly bias the fused posterior. Removing $S\_3$ causes the largest AP drop in Table 2, confirming this as the primary driver of LGRL's advantage.
>
> Controlled ablations (50% PER+LMR):
>
> | Variant                   | Corel5k AP      | Pascal07 AP     |
> | - | - | - |
> | SIP-style shared PoE      | 0.425 ($-$9.3%) | 0.557 ($-$8.2%) |
> | SIP-style feature-only IB | 0.453 ($-$3.4%) | 0.591 ($-$2.6%) |
> | Full LGRL                 | **0.469**       | **0.607**       |
>
> ### W2: Significance — AP gains and high-missingness realism
>
> **An absolute AP gain of 1–2 points is already considered large by multi-label standards**: AP aggregates precision across all positive labels' ranks, so even a single percentage point gain reflects consistently better ranking of relevant labels across potentially hundreds of categories. As shown in Table 1, LGRL surpasses the second-best baseline by **5.5, 4.5, and 3.9 AP points** on Corel5k, Pascal07, and Mirflickr at 50% PER+LMR, and consistently outperforms **all four most recent top-venue methods published in 2025–2026** by a clear margin, as further detailed in the comparison below.
>
> | Method   | Venue        | Corel5k AP | Pascal07 AP | Mirflickr AP |
> |- | - | - | - | - |
> | RANK     | TPAMI 2025   | 0.427      | 0.555       | 0.602        |
> | DCSI     | TPAMI 2026   | 0.436      | 0.559       | 0.619        |
> | URDF     | TPAMI 2025   | 0.446      | 0.565       | 0.615        |
> | TDLSR    | NeurIPS 2025 | 0.450      | 0.590       | 0.631        |
> | SIP      | ICML 2024    | 0.414      | 0.552       | 0.603        |
> | **LGRL** |              | **0.469**  | **0.607**   | **0.653**    |
>
> **90% missingness is realistic.** In CheXpert (224K images), approximately 80% of studies contain only a frontal view and lack lateral acquisitions entirely, directly corresponding to a real-world view missing rate of 80–90%. Besides, **this setting is standard in the field**, as MTD (Liu et al., NeurIPS 2024), SIP (Liu et al., ICML 2024), RANK (Liu et al., TPAMI 2025), and TDLSR (Li et al., NeurIPS 2025) all evaluate at 90%. Under this condition, existing methods collapse while LGRL remains robust. From 50% to 90% on Corel5k, SIP drops by **41.5%** and MTD by **43.5%**, whereas LGRL drops by only **23.2%**. At 90% missingness, LGRL achieves AP=0.360, **exceeding SIP by 48.8%**.
>
> **Minor:** "Ave.R" will be harmonized to "AVE".

---

> > ### Author Rebuttal · Reviewer_tyRd · 2026-04-02
> >
> > I thank the authors for their response.  I appreciate the clarifications.  The authors have sufficiently addressed my concerns.  I will update my score.

---

> > > ### Author Response · Authors · 2026-04-06
> > >
> > > Thanks again for your feedback and for updating your score. We are glad the clarifications on originality and significance were satisfactory and will add the corresponding content in the camera-ready. We sincerely thank you for your effort in evaluating our submission.

---

### Official Review · Reviewer_ERpE · 2026-03-13

**Soundness:** 3
**Presentation:** 3
**Significance:** 3
**Originality:** 3
**Overall Recommendation:** 4
**Confidence:** 3

**Summary:**

This paper studies incomplete multi-view multi-label classification and proposes a method called LGRL. A main part of the idea is to use label semantics more directly in representation learning, bottleneck regularization, and multi-view fusion. The model combines a semantic-guided prior, a label-aware bottleneck objective, and a category-specific fusion scheme. Experiments on six benchmarks and two real application scenarios show generally better results than previous methods.

**Compliance With Llm Reviewing Policy:**

Affirmed.

**Key Questions For Authors:**

1. For Eq. (1), I was not fully sure why the prototype distribution needs to be learned separately when b_c is already a learnable embedding. What extra flexibility does this give? Also, why was softplus used here?
2. I was not fully sure how reliable the approximation behind the feature-consistency term is in practice. How sensitive is the method to this approximation and to the related loss weights?
3. Since the method does posterior estimation and fusion at the category level, how do the runtime and memory costs scale when the number of labels becomes large?
4. I also wonder whether the experimental comparison could be strengthened with more recent closely related baselines, or at least with a clearer explanation of the baseline selection and tuning protocol.

**Limitations:**

The paper does not explicitly talk about its limitations. It has a short impact statement, but I think it would be better if the authors discussed the main assumptions of the method, where it might fail, and how it scales in practice.

**Strengths And Weaknesses:**

**[Strength]**

1. The main idea is well motivated, especially in how the paper uses label semantics throughout the framework rather than only at the final prediction stage.
2. The category-specific fusion also seems like a sensible design choice, particularly when some views are missing and different labels may rely on different evidence.
3. The experiments are fairly broad, with multiple datasets, missing-view settings, and ablations, so the empirical support is reasonably solid.


**[Weakness]**

1. I was not fully convinced by the justification for some of the objective design choices, especially the approximation used for the feature-consistency term.
2. The full framework seems fairly complex and potentially expensive because of the category-wise posterior and fusion design, but the paper does not quantify the extra computational cost.
3. Although the empirical results are strong overall, I think the paper would be more convincing with stronger comparisons to recent closely related methods.

---

> ### Author Rebuttal · Authors · 2026-03-31
>
> We thank Reviewer ERpE for recognizing the soundness and the sensibility of the category-specific fusion design. We address all concerns below.
>
> **W1&Q2: Feature-consistency approximation.**
>
> The derivation proceeds in three stages from $\min I(\mathbf{x}^{\sim(v)}; \mathbf{z}^{(v)} \mid \mathbf{x}^{(v)})$:
>
> **Stage 1 (exact, data processing inequality):** $I(\mathbf{x}^{\sim(v)}; \mathbf{z}^{(v)} \mid \mathbf{x}^{(v)}) \leq I(\mathbf{x}^{\sim(v)}; \mathbf{z} \mid \mathbf{x}^{(v)}) = \mathbb{E}\_{p({\mathbf{x}})} D\_{\mathrm{KL}}(p(\mathbf{z}|{\mathbf{x}}) | p(\mathbf{z}|\mathbf{x}^{(v)}))$.
>
> **Stage 2 (strict upper bound):** Introducing $q_\phi(\mathbf{z}|\mathbf{x}^{(v)})$ and dropping the non-negative term $D_{\mathrm{KL}}(p(\mathbf{z}|\mathbf{x}^{(v)}) | q_\phi(\mathbf{z}|\mathbf{x}^{(v)})) \geq 0$ yields a **provably valid** upper bound, conservative by construction.
>
> **Stage 3 (the approximation):** Replacing intractable $p(\mathbf{z}|{\mathbf{x}})$ with PoE variational approximation $q(\mathbf{z}|{\mathbf{x}})$. This is standard practice in multi-view VAE literature (MVAE, Wu & Goodman NeurIPS 2018; MMVAE, Shi et al. NeurIPS 2019) and is progressively aligned with the true posterior through joint optimization. Sensitivity to $\alpha$:
>
> | $\alpha$     | 0.1   | 0.5   | 1         | 2     | 5     |
> | ------------ | ----- | ----- | --------- | ----- | ----- |
> | Corel5k AP   | 0.465 | 0.467 | **0.469** | 0.468 | 0.464 |
> | Mirflickr AP | 0.649 | 0.651 | **0.653** | 0.652 | 0.648 |
>
> Stable within $[0.5, 2]$, confirming robustness.
>
> **W2&Q3: Computational cost.**
>
> LGRL introduces $\mathcal{O}(C \cdot d_z)$ additional space (approximately 150K parameters for $C$=291, $d_z$=512, less than 1.5% of the encoder stack) and $\mathcal{O}(C \cdot |\mathcal{V}| \cdot d_z)$ additional time per sample, both scaling linearly with $C$ in theory but sub-linearly in practice via GPU parallelism. Total runtime comparison (seconds, from Table 5):
>
> | Dataset   | $C$  | LGRL | MTD  | SIP  |
> | - | - | - | - | - |
> | Pascal07  | 20   | 591  | 1457 | 583  |
> | Mirflickr | 38   | 813  | 563  | 727  |
> | ESPGame   | 268  | 1738 | 4081 | 1605 |
> | Corel5k   | 260  | 1287 | 2521 | 1040 |
>
> LGRL remains substantially faster than MTD and LMVCAT across all datasets, and incurs only modest overhead compared to SIP and DICNet. For $C \gg 500$, prototype clustering or sparse attention would be natural approximations.
>
> **W3&Q4: Recent baselines.**
>
> To strengthen the empirical comparison, we benchmarked LGRL against four recently published methods from 2025–2026 (50% PER+LMR):
>
> | Method   | Venue        | Corel5k AP | Pascal07 AP |
> | -------- | ------------ | ---------- | ----------- |
> | RANK     | TPAMI 2025   | 0.427      | 0.555       |
> | DCSI     | TPAMI 2026   | 0.436      | 0.559       |
> | URDF     | TPAMI 2025   | 0.446      | 0.565       |
> | TDLSR    | NeurIPS 2025 | 0.450      | 0.590       |
> | **LGRL** | —            | **0.469**  | **0.607**   |
>
> LGRL outperforms all recent methods, including the strongest baseline TDLSR by +4.2% AP on Corel5k and +2.9% on Pascal07. We will add these results and baseline selection rationale to the camera-ready.
>
> **Q1: Why Label Encoder + softplus?** Direct parameterization of $\boldsymbol{\mu}_c$ is a linear mapping from label identity to the latent space, insufficient to capture the nonlinear geometric relationships needed to align prototypes with deep encoder representations. The shared MLP $h\_\mu$ enables nonlinear mapping, and its shared weights across all $\mathbf{b}_c$ implicitly regularize prototype geometry, encouraging label correlation structure to emerge without explicit co-occurrence supervision. Softplus ($\log(1+\exp(\cdot))$) ensures strict positivity for $\boldsymbol{\sigma}_c^2$, avoids numerical overflow of $\exp(\cdot)$, and is smooth everywhere, ensuring stable variance learning throughout training.
>
> **Limitations section.** We commit to adding explicit discussion of (1) scaling threshold for large $C$ and potential prototype clustering strategies; (2) graceful degradation under single-view availability, where Eq. 10 still yields a valid posterior; (3) the label conditional independence assumption; (4) potential underperformance under severe label imbalance.

---

> > ### Author Rebuttal · Reviewer_ERpE · 2026-04-02
> >
> > I thank the authors for the detailed rebuttal. As my primary concerns have been adequately addressed, I will maintain my positive score.

---

> > > ### Author Response · Authors · 2026-04-06
> > >
> > > Thanks again for your feedback and for maintaining your positive score. We are glad our responses addressed your concerns and will incorporate all suggestions in the camera-ready. We sincerely thank you for your effort in evaluating our submission.

---

### Decision · Program_Chairs · 2026-04-30

**Decision:**

Accept (regular)

**Comment:**

This paper proposes a well-motivated label-guided framework for incomplete multi-view multi-label classification, with meaningful technical novelty in its label-aware information bottleneck design and category-specific Bayesian fusion mechanism. The reviewers found the theoretical derivations rigorous, the empirical evaluation comprehensive, and the practical relevance clear. The authors’ rebuttal adequately addressed concerns regarding scalability, approximation quality, and differentiation from prior work. Overall, I find the paper technically sound and recommend acceptance.